# ESCAPING MODEL COLLAPSE VIA SYNTHETIC DATA VERIFICATION: NEAR-TERM IMPROVEMENTS AND LONG-TERM CONVERGENCE

**Bingji Yi**[*]
Independent Researcher
yibingji@gmail.com

**Qiyuan Liu**[*]
Department of Statistics
University of Chicago
qiyuanliu@uchicago.edu

**Yuwei Cheng**
Department of Statistics
University of Chicago
yuweicheng@uchicago.edu

**Haifeng Xu**
Department of Computer Science
University of Chicago
haifengxu@uchicago.edu

## ABSTRACT

Synthetic data has been increasingly used to train frontier generative models. However, recent studies raise key concerns that iteratively retraining a generative model on its self-generated synthetic data may keep deteriorating model performance, a phenomenon often coined *model collapse*. In this paper, we investigate ways to modify the synthetic retraining process to avoid model collapse, and even possibly help reverse the trend from collapse to improvement. Our key finding is that by injecting information through an external synthetic data verifier, whether a human or a better model, synthetic retraining will not cause model collapse. Specifically, we situate our theoretical analysis in the fundamental linear regression setting, showing that verifier-guided retraining can yield near-term improvements, but ultimately drives the parameter estimate to the verifier's "knowledge center" in the long run. Our theory further predicts that, unless the verifier is perfectly reliable, these early gains will plateau and may even reverse. Indeed, our experiments across linear regression, Variational Autoencoders (VAEs) trained on MNIST, and fining-tuning SmolLM2-135M on the XSUM task confirm these theoretical insights.

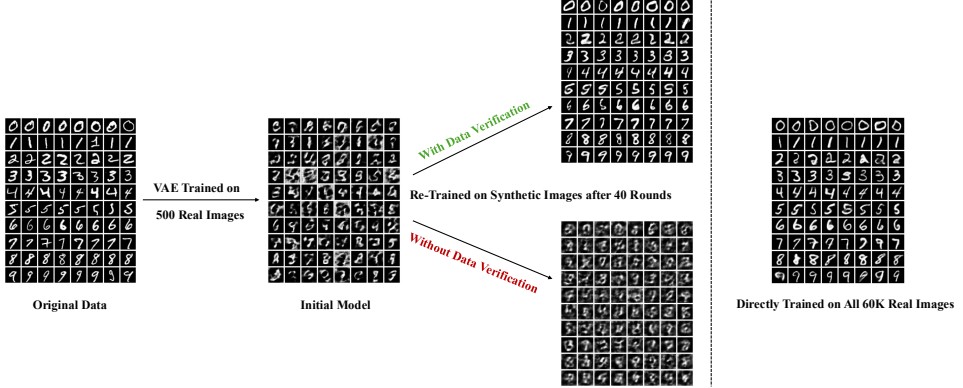

Figure 1: **Iterative VAE Retraining on MNIST. Left:** Original MNIST images (real data). **Middle:** Samples from a VAE trained on 500 real images. **Right:** Samples after 40 rounds of synthetic retraining. The *top branch* (green) uses verifier-filtered synthetic data, producing clearer and more realistic digits; the *bottom branch* (red) retrains without verification, leading to severe degradation and mode collapse. The final column shows a VAE trained on all 60K real images (upper bound on quality).

---

[*]Equal contribution.

# 1 INTRODUCTION

The use of synthetic data has gained significant traction due to its ability to reduce data collection costs and enhance privacy protection, with applications in computer vision (Wood et al., 2021), healthcare (Azizi et al., 2021; Santangelo et al., 2025), finance (Potluru et al., 2023) and recently large language models (LLMs) (Chen et al., 2024). A growing body of work has demonstrated that training with synthetic data can improve model performance in various applications such as image recognition (He et al., 2023; Tremblay et al., 2018), image generation (Doersch & Zisserman, 2019; Shrivastava et al., 2017; Tian et al., 2023) and language generation (Gunasekar et al., 2023; Guo et al., 2024; Zelikman et al., 2022). However, recent studies caution that recursively training models on synthetic data alone can lead to a degradation of quality, a phenomenon often coined *model collapse* (Shumailov et al., 2024; Dohmatob et al., 2024a; 2025; 2024b; Alemohammad et al., 2024; Gerstgrasser et al., 2024). This contrast between empirical success and pessimistic research findings gives rise to a natural question about what might have caused the discrepancy.

In response to the above question, an important practical observation is that synthetic data are rarely used in raw form in the aforementioned applications. Instead, practitioners often apply filtering steps to remove low-quality synthetic samples before retraining. For example, in natural language generation, synthetic sentences are often screened using grammar checkers or LLM-as-a-judge pipelines (Zheng et al., 2023; Gu et al., 2024); in computer vision, synthetic images can be filtered using pretrained discriminators or human annotation (He et al., 2023); in recommendation and preference learning, synthetic feedback is often validated against external heuristics or known user signals (Tu et al., 2025; Iskander et al., 2024; Lupidi et al., 2024; Lampis et al., 2023; Zhang et al., 2024). Common across all these approaches is the use of a knowledgeable "discriminator" (henceforth the *verifier* ) – whether a machine or human – that evaluates and filters out low-quality candidate synthetic samples (i.e., those not passing the discriminator's screening). This observation naturally raises the following research question:

> *Does verifier-based filtering of synthetic data contribute to the observed empirical success of model improvements, and does it prevent model collapse in the long run?*

There has been a growing body of work studying the mechanisms of *model collapse*, including theoretical analyzes that are often examined in the context of classic parameter estimation problems (Dohmatob et al., 2024a; 2025; Gerstgrasser et al., 2024; Dey & Donoho, 2024; Xu et al., 2025; Suresh et al., 2025). However, these works have all assumed the use of synthetic data *without* filtering. Only few recent works start to examine how filtered synthetic data affects the performance of iterative retraining, but under idealized assumptions such as access to a perfectly correct verifier (Amin et al., 2025) or highly structured errors in synthetic data (i.e., i.i.d. noise added to binary labels (Feng et al., 2025)). A realistic and instructive framework for analyzing synthetic retraining of generative models remains still poorly understood. In this paper, we further push the boundary of this important agenda, and examine iterative retraining under imperfect verifiers that filter out low-quality synthetic data based on their (possibly biased) knowledge. We refer to this process as *verifier-based synthetic retraining* for convenience. Specifically, we seek principled understandings about the empirically observed short-term successes of verifier-based synthetic retraining and analysis about its long-term convergence under iterative retraining.

**Our contributions.** We start from theoretical investigations and analyze verifier-based synthetic retraining with *verified* synthetic data on the foundational linear regression model—a canonical setting that has become central to the study of model collapse (e.g., (Dohmatob et al., 2024a; 2025; Gerstgrasser et al., 2024; Zhu et al., 2025; Garg et al., 2025)).[1] We then verify our theoretical insights through thorough empirical studies in real-world generative settings. Our main contributions are summarized as follows:

- *Does verified synthetic data improve retraining and, if so, under what conditions?* We show that it indeed can, provided the right conditions are met. Through a new form of bias-variance trade-off under data filtering, we characterize the regimes in which verifier-based synthetic retraining leads to strict model improvement, rather than degradation, in the short

---

[1]The theoretical analysis of (Feng et al., 2025) is also based on linear models, specifically, linear classifiers.

term (Theorem 3.1). The conditions we identify highlight the mixed effect of synthetic sample size, the verifier's bias and selectivity during filtering, yielding practical insights regarding *when* verification of synthetic data is beneficial.

- *Can the retraining improvement in the short term be sustained in the long run?* Our theoretical analysis shows that the answer is *No* – unless the verifier has no bias. Formally, we show that verifier-based synthetic retraining will converge to the verifier's knowledge center in the long term (Theorem 4.1). This result reveals how the verifier's quality affects the asymptotic dynamics of iterative retraining. Notably, while verifier selectivity influences short-term performance, it does not change the long-run converging point, though it does affect its convergence rate.

- *Do the insights from the above theoretical analysis apply to real generative models empirically?* Indeed, both simulations of linear regression settings and training Variational Autoencoders (VAEs) on the real-world MNIST data demonstrate that our theoretical insights align with observed training dynamics in these settings. For example, Figure 1 shows how the generative model of VAE can start from a poor model trained on a small number of 500 images and gradually improve from being retrained on filtered synthetic imaged generated by itself, until after 40 iterations of retraining generating sharp images that are visually comparable to a VAE trained directly on the entire MNIST dataset. More quantitative analysis of our empirical studies can be found in Section 5.

## 1.1 RELATED WORK

**Understanding and mitigating model collapse.** Recent work shows that heavy reliance on synthetic data in iterative training can cause *model collapse*—the degradation of performance when a model is repeatedly retrained on its own synthetic outputs (possibly mixed with real data).[2] Empirical evidence supports this phenomenon: Shumailov et al. (2024) show that recursive training on unfiltered synthetic data induces distribution shift and mode collapse, while Dohmatob et al. (2025) find that even small synthetic proportions can harm performance. In linear settings, Dohmatob et al. (2024a) analyze collapse mechanisms explicitly, and Dohmatob et al. (2024b) connect degradation to altered neural scaling laws.

To mitigate collapse, prior work broadly explores three strategies. First, accumulating data or gradually increasing the synthetic dataset size across iterations can suppress noise and bound errors (Gerstgrasser et al., 2024; Dey & Donoho, 2024; Xu et al., 2025; Kazdan et al., 2025; Barzilai & Shamir, 2025). Second, mixing synthetic data with real data stabilizes retraining (Bertrand et al., 2024; Fu et al., 2024; 2025), as performance progressively degrades without sufficient fresh real data (Alemohammad et al., 2024). Recent studies have even derived optimal mixing ratios to maximize this stabilizing effect (He et al., 2025; Garg et al., 2025). Finally, algorithmic interventions, such as the token re-sampling procedures proposed by Zhu et al. (2025), offer alternative pathways to avoid collapse.

Unlike prior work that relies on unfiltered synthetic data, our framework incorporates an external verifier to remove low-quality samples. Such verifiers may be human annotators or stronger teacher models. Filtering is widely used in iterative retraining and has shown empirical success in preventing model degradation and even improving performance (He et al., 2023; Tian et al., 2023; Guo et al., 2024; Zelikman et al., 2022; Zhang et al., 2024; Lampis et al., 2023; Haluptzok et al., 2023; Patwa et al., 2024). Motivated by this, we develop a principled understanding of when improvement is possible—namely, whether a generative model can leverage the verifier's feedback, embedded in the selected synthetic subset, to achieve sustained gains.

**Filtering and selecting synthetic data.** While a rich line of empirical work demonstrates that these filtering strategies can improve model performance, theoretical understanding about iterative retraining with filtered synthetic data remains largely unexplored, with only a few recent exceptions. Amin et al. (2025) assume a strong, reliable quality function and focus on how an external labeler aids learning under this fixed filtering mechanism. Feng et al. (2025) study a classification problem and identify a sharp phase transition. However, modeling synthetic data merely as noisy labels abstracts away the structural dependencies between features and labels inherent to true generative processes. Finally, Ferbach et al. (2024); Wei & Zhang considers a conceptually similar problem of

---

[2]There is no widely agreed-upon formal definition; see (Schaeffer et al., 2025) for discussion.

learning a discrete preference distribution from human feedback by using humans' preferred choices as a filtering strategy. In their population-level analysis, curating synthetic data via an external reward function forces the model distribution to converge to the highest-reward level set, maximizing expected reward but ultimately collapsing in diversity.

Similar to many of the aforementioned studies above, our theoretical analysis also focuses on linear models (Feng et al., 2025; Dohmatob et al., 2025; Gerstgrasser et al., 2024). However, our model allows inaccuracy of the verifier in terms of both bias and variance. Errors in the synthetic data primarily stem from the inaccuracy of the generative model itself rather than exogenous noise. We show that model's short-term performance varies smoothly with the verifier's bias, selectivity, and size of synthetic data, rather than exhibiting a sharp phrase transition from complete failure to perfect accuracy as in Feng et al. (2025). In the long run, the model's performance converges to the verifier's knowledge center whereas verifier's selectivity only affects convergence speed. Our results bridge short-term and long-term perspectives of iterative retraining, illustrating how varied verifier qualities give rise to distinct performances of iterative retraining.

**Comparison with reward maximization frameworks.** While both use external feedback to evaluate generated data, our approach substantially differs in various aspects from reward-maximization frameworks such as preference matching (Ferbach et al., 2024; Wei & Zhang) and RLVR (Guo et al., 2024; Yu et al., 2025). We provide a detailed comparison of these paradigms in Appendix B.

## 2 MODELING VERIFIER-BASED SYNTHETIC RETRAINING: THE LINEAR REGRESSION CASE

In this section, we formalize our model of iterative retraining with verified synthetic data, coined *verifier-based synthetic retraining* for convenience. Following recent works in this space (Dohmatob et al., 2024a; Gerstgrasser et al., 2024; Garg et al., 2025; Zhu et al., 2025; Garg et al., 2025), we focus on the foundational linear regression setting where the objective is to estimate a high-dimensional coefficient vector $\theta^\star$ in the following linear model

$$y = x^\top \theta^\star + \xi,$$

where $\xi \sim \mathcal{N}(0, \sigma^2)$, $x \in \mathbb{R}^p$, and $\theta^\star \in \mathbb{R}^p$ is the unknown parameter of interest. We use the standard Mean Squared Error (MSE), i.e., $\text{MSE}(\hat{\theta}) = \mathbb{E}_\xi \|\hat{\theta} - \theta^\star\|^2$, to evaluate estimators.

**Modeling the verifier and data filtering rule.** Suppose we have access to a verifier that possesses prior knowledge of $\theta^\star$, modeled by a knowledge set. Specifically, the verifier's knowledge is described by a spherical ball:

$$B_r(\theta_c) := \big\{ \theta \in \mathbb{R}^p : \|\theta - \theta_c\| \leq r \big\},$$

with fixed center $\theta_c$ and radius $r$. We assume this knowledge set indeed contains the true parameter, i.e., $\theta^\star \in B_r(\theta_c)$, but the true parameter $\theta^\star$ is unknown. The verifier does not reveal $\theta_c$ or $r$ directly (see modeling motivations below). Instead, it only provides binary feedback indicating whether a given (real or synthetic) data point $(x_i, y_i)$ is consistent with the knowledge $\theta^\star \in B_r(\theta_c)$ or not. Specifically, the verifier outputs *Yes* if

$$|y_i - x_i^\top \theta_c| \leq r\|x_i\| + \sigma_c, \tag{1}$$

and *No* otherwise. Here $\sigma_c$ is a constant related to the verifier's capability. This *filtering rule* is motivated by the following bound on expected errors: $\mathbb{E}\big[|y_i - x_i^\top \theta_c|\big] = \mathbb{E}\big[|x_i^\top(\theta^\star - \theta_c) + \xi_i|\big] \leq r\|x_i\| + \mathbb{E}|\xi_i| = r\|x_i\| + \sqrt{\frac{2}{\pi}}\sigma$. Since the true $\sigma$ might be unknown in practice, $\sigma_c$ serves as an estimate of the true $\sigma$.

We refer to $\Delta = \|\theta^\star - \theta_c\|$ as the *bias* of the verifier, whereas $r$ captures the *selectivity* of the verifier – the smaller $r$ is, less likely the verifier accepts a data point $(x_i, y_i)$. The verifier only needs to provide Yes/No answers based on the above selection rule in equation 1, but does not needs to know the parameter $\theta_c, r$ of the knowledge set. The motivation of this modeling primarily comes from practice, as explained below.

**Motivation of binary feedback from verifiers.** We adopt the binary feedback from verifiers mainly for practical reasons. In practice, eliciting simple yes/no feedback is far less noisy and more cost-effective than asking verifiers to directly specify $\theta_c$ or $r$. Indeed, in real applications verifiers may not even know these quantities explicitly, which would correspond to model parameters if the verifier is a stronger teacher model or how the human reasons if the verifier is a human. This model choice is also aligned with the widely adopted comparison-based feedback in reinforcement learning from human feedback (RLHF) (Ouyang et al., 2022). Such binary feedback has become a standard approach in preference alignment for large language models, where LLM raters and human evaluators provide pairwise or accept/reject judgments that effectively guide learning at scale (Wettig et al., 2024). Although simple, our theory and empirical evaluations both show that this *single bit* of information for each sample can successfully be injected into the retraining process to improve models.

**Synthetic Retraining with Verifier-based Filtering** We begin with an initial set of real data $(X^0, Y^0)$, where $X^0 \in \mathbb{R}^{n_0 \times p}$ and $Y^0 \in \mathbb{R}^{n_0}$. The initial estimator $\hat{\theta}^0$ is obtained via Ordinary Least Squares (OLS) [3] $\hat{\theta}^0 = (X^{0\top} X^0)^{-1} X^{0\top} Y^0$. We then proceed with iterative synthetic retraining via the *generate–verify–retrain* procedure outlined in Figure 2, the rigorous retraining Scheme 2 and Algorithm 3 are provided in Appendix C.

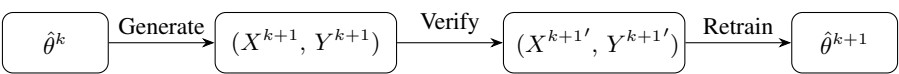

Figure 2: Generate-Verify-Retrain pipeline.

Since learning proceeds through the conditional $Y^k \mid X^k$, synthetic retraining requires specifying the covariate design $X^k$; labels $Y^k$ are then generated conditionally via the model under verifier constraints. In principle, one could construct $X^k$ arbitrarily; however, for mathematical clarity, below we describe a targeted though arguably natural design. In particular, we choose to align the synthetic covariates with a fixed orthonormal set $\{v_1, \ldots, v_p\}$ and construct $X^k$ in a block-structured form by repeating each $v_j^\top$ as rows:

$$X^k = (\ \underbrace{v_1, \ldots}_{\text{copies of } v_1}, \ \underbrace{v_2, \ldots}_{\text{copies of } v_2}, \ \ldots, \ \underbrace{v_p, \ldots}_{\text{copies of } v_p} \ )^\top. \tag{2}$$

After verifier filtering, each orthogonal direction $v_j$ retains exactly $n_k$ samples with $n_0 \le pn_1 \le pn_2 \le \cdots$.

Notably, the estimation of parameter $\hat{\theta}^{k+1}$ using only synthetic data from the model with $\hat{\theta}^k$, though with filtering, leads to a Markovian transition $\hat{\theta}^k \mapsto \hat{\theta}^{k+1}$. The above block design essentially helps "diagonalize" the transition operator $\hat{\theta}^k \mapsto \hat{\theta}^{k+1}$. The conceptual benefit of this covariance design choice is that we remove the rotational variability that arbitrary designs would introduce across iterations and decouple the dynamics along orthogonal directions. In practice, this design mirrors curating data along approximately orthogonal latent spaces or topics (e.g., topical axes like politics, sports, mathematics). However, the choice of covariant $X^k$ is not unique: alternatives (e.g., canonical basis, isotropic random directions) can yield similar qualitative conclusions, with potentially different constants or rates. We expect our theoretical insights to generalize to any reasonable design of the covariant $X^k$, though the rigorous proofs may be less tractable for some designs.

## 3 ON THE NEAR-TERM IMPROVEMENT UNDER SYNTHETIC RETRAINING

This section investigates the verifier's role in synthetic retraining: *does it help, when does it help, and why does it help?* We focus on one round and show that verifier-guided retraining can improve performance under mild assumptions. The underlying mechanism is a verifier-induced bias-variance trade-off: filtering synthetic data *reduces variance* but may *introduce bias*.

---

[3]For ease of presentation, we assume $\mathrm{Rank}(X^0) = p$. If $\mathrm{Rank}(X^0) < p$, all our results are equally applicable by working in the subspace of $X^0$.

## 3.1 SOURCE OF IMPROVEMENT: BIAS–VARIANCE TRADE-OFF

To understand *why* verifier-based retraining can improve upon the initial estimator $\hat{\theta}^0$, we must examine the fundamental bias–variance trade-off introduced by the filtering process. The initial estimator $\hat{\theta}^0$ is unbiased but suffers from high variance due to the limited real sample size $n_0$. When we generate synthetic data and apply the verifier, we effectively discard inconsistent samples. This filtering reduces the estimation variance. However, because the verifier itself may be imperfect, this filtering injects a systematic bias. Therefore, synthetic retraining yields a strictly better model precisely when the variance reduction achieved by filtering outweighs the injected bias and the sampling noise of the synthetic data itself.

## 3.2 CHARACTERIZING IMPROVEMENT IN ONE-ROUND RETRAINING

The following theorem rigorously quantifies this trade-off for the linear regression model, demonstrating exactly *when* the one-step estimator $\hat{\theta}^1$ improves or degrades upon the initial baseline. By characterizing the MSE of $\hat{\theta}^1$, it reveals how synthetic sample size, verifier bias, and verifier selectivity determine the final outcome.

**Theorem 3.1.** *Let $\{\mu_j\}_{j=1}^p$ denote the singular values of $X^0$, and assume each of them satisfies $\mu_j = \Omega(\sqrt{n_0})$.[4] Then there exist constants $m_{1,j}, m_{3,j} \in \mathbb{R}$ and $m_{2,j} \in (0,1)$ for $j = 1, \ldots, p$, as well as a constant $L > 0$ such that:*

$$\frac{1}{\sigma^2}MSE(\hat{\theta}^1) = \sum_{j=1}^p \left( \underbrace{\frac{m_{2,j}}{n_1}}_{\text{Synthetic Variance}} + \underbrace{m_{1,j}^2 + \frac{m_{1,j}m_{3,j} + m_{2,j}^2}{\mu_j^2}}_{\text{Verification Error}} \right) + \mathcal{O}\left( n_0^{-4/3} \right) \qquad (3)$$

*holds with probability at least $1 - p \exp\left( -Ln_0^{1/3} \right)$, where $n_1$ denotes the post-verification sample size.*

While the explicit forms of the constants are deferred to Appendix C, their roles are highly intuitive: $m_{1,j}$ and $m_{3,j}$ capture the directional bias between the verifier's knowledge center $\theta_c$ and the ground truth $\theta^*$ along the $j$-th singular direction (vanishing if $\theta_c = \theta^*$), while $m_{2,j} < 1$ quantifies the variance reduction along that direction. Theorem 3.1 mathematically guarantees when verifier-guided retraining improves the model. Since the scaled baseline error is $\frac{1}{\sigma^2}\text{MSE}(\hat{\theta}^0) = \sum_{j=1}^p \mu_j^{-2}$, we can directly compare it to Equation 3. When the verifier is highly accurate ($m_{1,j}, m_{3,j} \approx 0$), the verification error term becomes dominated by $\sum_{j=1}^p m_{2,j}^2/\mu_j^2$. Because $m_{2,j} < 1$, this verification error is strictly smaller than the real-data error. Thus, whenever the verified synthetic sample size $n_1$ is sufficiently large to drive the synthetic variance down, $\text{MSE}(\hat{\theta}^1)$ strictly improves upon the baseline.

This result highlights why verifier-based retraining is practically useful: in modern machine learning systems where real data collection is costly but generative models or simulators are available, a moderately accurate verifier can filter synthetic samples to effectively amplify limited real-world evidence and substantially reduce estimation error. Conceptually, this offers a sharp departure from classical model collapse literature, which typically models iterative synthetic data purely as a variance-inflating noise source (Shumailov et al., 2024; Alemohammad et al., 2024; Dohmatob et al., 2024a). Here, we prove that *verification transforms synthetic data into a variance-reducing resource*, provided the verifier's bias is sufficiently controlled. As we will demonstrate empirically in Section 5, this mechanism is not confined to the linear model; it manifests clearly in complex models such as VAEs and LLMs.

## 4 ITERATIVE RETRAINING CONVERGES TO THE VERIFIER'S KNOWLEDGE CENTER

Having established that a single round of verifier-based retraining can improve estimation through a bias–variance trade-off, a natural question arises: *can this improvement be sustained over mul-*

---

[4]That is, each dimension is well-represented in the original data. This holds easily when, e.g., the feature data is drawn i.i.d. from a full-rank distribution.

*tiple rounds, and what is the eventual outcome?* To contextualize our long-term analysis within the broader literature on model collapse, we first formalize these widely discussed empirical phenomena within our linear regression setting. Specifically, we define **Model Degradation/Collapse** as $\limsup_{k\to\infty} \mathrm{MSE}(\hat{\theta}^k) > \mathrm{MSE}(\hat{\theta}^0)$, and **Model Improvement** as $\limsup_{k\to\infty} \mathrm{MSE}(\hat{\theta}^k) < \mathrm{MSE}(\hat{\theta}^0)$.

Our key finding is that both behaviors can occur in long-term iterative retraining. The outcome depends critically on three factors: the growth rate of synthetic data, the verifier's bias, and the verifier's capability (i.e., its ability to reduce variance). Over time, iterative retraining injects increasingly more verifier knowledge into the estimator, while the contribution from the original data gradually decays. As a result, the verifier and the generative model family eventually dominate the limit behavior, driving the estimator $\hat{\theta}^k$ toward a fixed point, which corresponds to the verifier's knowledge center $\theta_c$.

This dynamic gives rise to three distinct phases of long-term behavior: **(1) Unbiased verifier:** If the verifier is unbiased (i.e., $\theta_c = \theta^\star$), iterative retraining yields continuous improvement and the estimator converges to the true parameter. **(2) Mildly biased verifier:** With small bias, iterative retraining can improve performance in the short term by reducing variance, but performance eventually plateaus or deteriorates as verifier bias accumulates. **(3) Strongly biased verifier:** With large bias, iterative retraining leads to degradation and may even cause collapse in the limit. Among these, the mildly biased case is the most practically relevant. It highlights a cautionary message: while synthetic retraining can initially boost accuracy, a perfectly unbiased verifier is unrealistic; consequently, this inherent bias will ultimately prevent sustained improvement.

Formally, the following theorem characterizes the long-term behavior of the estimator $\hat{\theta}^k$ in linear regression under verifier-based synthetic retraining.

**Theorem 4.1.** *There exist synthetic retraining processes (e.g., Algorithm 3) and a constant $0 < \rho < 1$ such that:*

$$\mathbb{E}\|\hat{\theta}^k - \theta_c\|^2 \le \rho^{2k}\mathbb{E}\|\hat{\theta}^0 - \theta_c\|^2 + p\sigma^2 \sum_{j=0}^{k-1} \frac{\rho^{2(k-j)-1}}{n_j}. \tag{4}$$

*where $n_1 \le n_2 \le \cdots \le n_k \cdots$ denote the number of verified synthetic samples per direction at each iteration. In particular, if $\lim_{k\to\infty} n_k = \infty$, then $\lim_{k\to\infty}\mathbb{E}\|\hat{\theta}^k - \theta_c\|^2 = 0$.*

The proof of Theorem 4.1 is provided in Appendix C, utilizing concentration bounds and supermartingale inequalities to establish convergence. Here we focus on the main intuition and highlight the key novelty of our analysis.

The central observation to establish Theorem 4.1 is that the iterative retraining procedure (see Figure 2) induces a *Markov process*: the next state $\hat{\theta}^{k+1}$ depends only on the current state $\hat{\theta}^k$. Formally, the update can be expressed as

$$\hat{\theta}^{k+1} = T(\hat{\theta}^k) + \eta_{k+1}, \tag{5}$$

where $T(\cdot)$ is a deterministic mapping determined by verifier filtering, and $\eta_{k+1}$ is a sub-Gaussian noise term due to the randomness of synthetic samples at iteration $k+1$. Crucially, we show that $T(\cdot)$ is a *contraction mapping*, and that the variance of the noise decays at the rate $\mathrm{Var}(\eta_{k+1}) \asymp 1/n_{k+1}$. This perspective allows us to view equation 5 as a discretized stochastic differential equation (SDE). As $n_k \to \infty$, the noise term vanishes and the dynamics are dominated by the deterministic contraction $T(\hat{\theta}^k)$, which drives the recursion toward its fixed point—the verifier's knowledge center $\theta_c$. The presence of the verifier is therefore *essential*: it is precisely what transforms the update rule into a contraction, guaranteeing convergence.

By contrast, in prior work on model collapse without a verifier (e.g., Gerstgrasser et al. (2024); Xu et al. (2025)), the update reduces to the identity mapping. In that case, increasing the synthetic sample size can suppress noise accumulation and ensure bounded error (i.e., $\mathrm{MSE}(\hat{\theta}^k) < \infty$), but there is no contraction and hence no convergence or sustained improvement. The knowledge extracted from the verifier is precisely what elevates $T(\cdot)$ beyond an identity mapping. Our analysis is the first to formally show that the verifier fundamentally alters the long-term dynamics: it continuously injects knowledge, iteration by iteration, so that the estimator moves closer to $\theta_c$ over time.

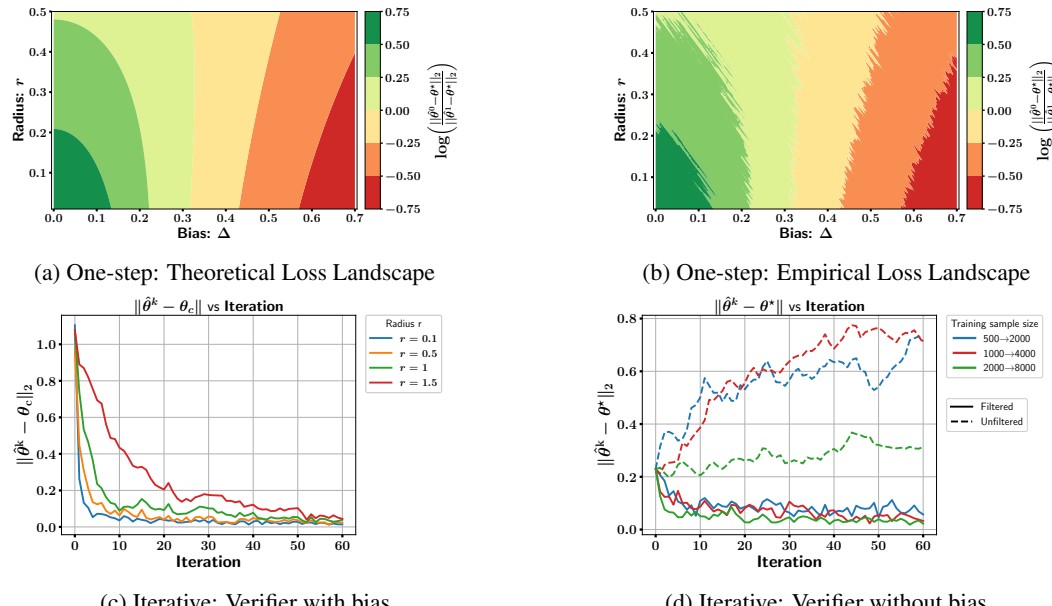

(a) One-step: Theoretical Loss Landscape

(b) One-step: Empirical Loss Landscape

(c) Iterative: Verifier with bias

(d) Iterative: Verifier without bias

Figure 3: **Top:** Error changes of the one-step retraining estimator $\hat{\theta}^1$ versus estimator $\hat{\theta}^0$ only using original real data, measured by $\log(\frac{||\hat{\theta}^0-\theta^\star||}{||\hat{\theta}^1-\theta^\star||})$: theory's prediction (a) and empirical comparisons (b). **Bottom:** Iterative synthetic retraining over 60 rounds with a biased verifier ($\Delta = 1$) and an unbiased verifier ($\Delta = 0$).

This contribution also clarifies a common misconception: even with a perfect verifier ($\theta_c = \theta^\star$) and infinitely many synthetic samples in one iteration, convergence cannot occur in a single step. As shown in Theorem 3.1, while infinite samples remove the synthetic variance term, the verification error term persists. Thus, convergence requires the *iterative* action of the verifier, which gradually aligns the estimator with the truth.

## 5 EXPERIMENTS

In this section, we validate our theoretical predictions across three settings: a *linear regression simulation* that is consistent with our analytical assumptions, alongside *Variational Autoencoders (VAEs) on MNIST* and fine-tuning a pretrained SmolLM2-135M (Allal et al., 2025) on a large-scale news summarization task, which together illustrate practical behavior under iterative retraining and filtering. Across all settings, the empirical results closely align with our theoretical predictions. Experimental code used to generate the results is publicly available at https://github.com/liuqiyuanhhh/Verified-Synthetic-Data.

### 5.1 LINEAR REGRESSION ON SIMULATED DATA

**Setting.** We consider the linear model $y = x^\top \theta^\star + \xi$, with $\xi \sim \mathcal{N}(0, 1)$, $\theta^\star \in \mathbb{R}^p$, and $x \in \mathbb{R}^p$. We first fit an OLS estimator on the real dataset $(X^0, Y^0)$, and then perform multiple rounds of synthetic retraining, where the synthetic covariate design is aligned with the right singular vectors of $X^0$.

**One-step Synthetic Retraining.** Figures 3a and 3b empirically validates Theorem 3.1 by comparing the real-data estimator $\hat{\theta}^0$ with the one-step synthetic estimator $\hat{\theta}^1$, where color regions reflect different levels of error reduction, measured by $\log(\frac{||\hat{\theta}^0-\theta^\star||}{||\hat{\theta}^1-\theta^\star||})$. Theoretical predictions (Figure 3a) align closely with empirical results (Figure 3b), validating the sharpness of our bounds. Using 100 real and 800 synthetic samples, we set $\theta^\star = \mathbf{1}_8$ and the verifier center $\theta_c = \theta^\star + \Delta \cdot u$, where $u$ is a random unit vector and $\Delta$ controls bias magnitude. The radius $r$ (see equation 1) determines verifier selectivity. These results confirm that synthetic retraining outperforms the baseline under small bias (green

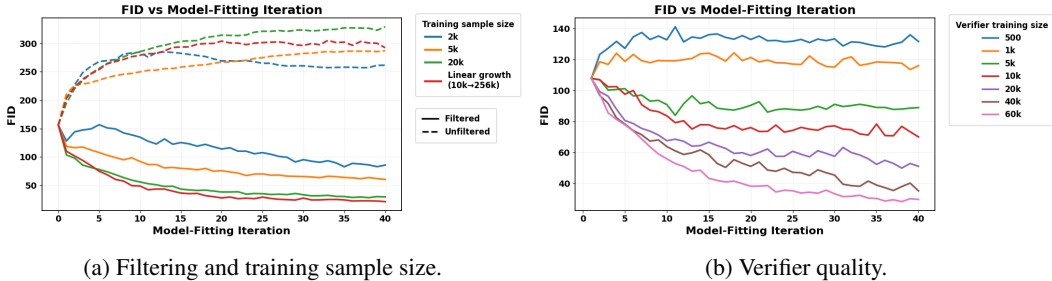

(a) Filtering and training sample size.

(b) Verifier quality.

Figure 4: FID results across retraining rounds. (a) Effect of filtering and retained sample size. (b) Effect of verifier quality, varied by training data size. Together, the plots highlight how both sample selection and verifier strength shape generative performance.

region) but degrades under excessive bias (red region), consistent with the predicted short-term bias–variance trade-off.

**Iterative Synthetic Retraining.** Figure 3c validates Theorem 4.1, showing that under a biased verifier ($\Delta = 1$), the retrained estimator converges to the verifier's knowledge center $\theta_c$ rather than the true parameter $\theta^\star$. In this 60-round experiment, sample size increases linearly from 100 to 5500, and results show that a more selective verifier (smaller $r$) accelerates convergence. Figure 3d presents the unbiased case ($\Delta = 0$), where verifier-based retraining consistently outperforms unfiltered baselines. Together, these findings illustrate how the verifier's contraction effect sustains error reduction and prevents model collapse. Additional experiments confirm that our conclusions are robust to random covariate designs (Appendix E.1) and different verifier shapes (Appendix E.2).

## 5.2 VARIATIONAL AUTOENCODERS (VAEs) ON MNIST

Extending beyond linear regression, we evaluate our theory on real-world image generation tasks using Variational Autoencoders (VAEs) on the MNIST dataset.

**Setting and evaluation metrics.** Specifically, we adopt Conditional VAEs (CVAEs) to leverage class conditioning and avoid verifier-induced imbalance; otherwise, easily generated digits would dominate the retained synthetic data. To test the bias–variance trade-off and verifier information injection, we initialize the CVAE with only 500 real images (a challenging *small dataset* scenario). A discriminator, trained on varying amounts of real data alongside an equal number of synthetic samples, serves as the verifier. It assigns a reality probability to each synthetic sample, from which we retain the top 10% per digit. Motivated by our one-step analysis, this 10% threshold optimally balances sample quality against diversity. The CVAE is iteratively retrained on verified data for 40 iterations until performance stabilizes. The synthetic sample size $n_k$ follows either a fixed or linear growth schedule. We evaluate generative performance using Fréchet Inception Distance (FID) (Heusel et al., 2017) and the evidence lower bound (ELBO) (Kingma & Welling, 2014). Appendices D and E.4 provide implementation details and initial sample size ablations.

**Results.** Since our verifier emphasizes perceptual realism over likelihood calibration, we report FID as the primary metric and defer qualitatively similar ELBO results to Appendix D. As shown in Figure 4a, synthetic retraining with a strong verifier (trained on 60K real images) yields rapid early FID improvement before plateauing, whereas unverified retraining (dashed curves) leads to severe degradation. This validates our theory: early gains arise from the short-term bias-variance trade-off (Theorem 3.1), and the verifier's contraction effect drives convergence to a fixed—though biased—point that substantially improves upon the initial CVAE (Theorem 4.1). The eventual plateau reflects two limitations: training the initial generator on only 500 real images inherently restricts overall diversity, and the standard MLP verifier lacks diversity-preserving mechanisms, introducing selection bias by disproportionately rejecting harder-to-generate modes. For reference, a baseline CVAE trained on 60K real images achieves 17.56 FID, whereas the best synthetic model reaches 21.17 after 40 iterations. Figure 4b further shows that with 20K synthetic samples per round, stronger verifiers (trained on more real data) produce larger FID improvements, while weaker ones cause early plateau or even degradation. Qualitative results are shown in Figure 1, with consistent MNIST-specific FID results reported in Appendix E.3.

## 5.3 LARGE-SCALE NEWS SUMMARIZATION

Extending beyond image generation, we evaluate our theory on natural-language tasks using the XSUM news-summarization dataset (Narayan et al., 2018).

**Setting and evaluation metrics.** We use the the pretrained `SmolLM2-135M` model (Allal et al., 2025) as our generator, first fine-tuning it on 12.5% of the XSUM training set for one epoch using full-parameter training. While following the base setup of Feng et al. (2025), we depart from their single-round evaluation by considering a *multi-iteration* generate–verify–retrain regime to capture how performance evolves over repeated cycles. Given the low-entropy nature of news summarization, we follow the common practice of employing greedy decoding for both generation and evaluation.

In each iteration, the model generates synthetic summaries for the training corpus, which are scored using ROUGE-1 against ground-truth references. Acting as an oracle verifier, we select the top 12.5% of these summaries to form the synthetic retraining dataset. The model is then retrained on this subset, and test ROUGE-1 scores are recorded after each iteration.

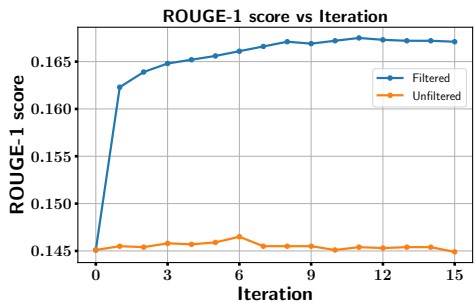

**Results.** Figure 5 reports ROUGE-1 scores across 15 rounds for both filtered and unfiltered retraining regimes, while all other experimental conditions remain identical. Under verifier filtering, performance improves monotonically during early iterations before eventually stabilizing. In contrast, the unfiltered baseline fluctuates around its initial score without meaningful improvement. These observed dynamics mirror our MNIST findings and theoretical predictions, demonstrating that our framework scales to natural-language tasks.

Figure 5: **ROUGE-1 score vs. iteration on the XSUM dataset.** Filtered retraining yields consistent early improvements, whereas unfiltered retraining shows no significant gain.

## 6 DISCUSSION

Our study provides a theoretical and empirical characterization of verifier-guided synthetic retraining. We show that the process yields *short-term gains* by reducing variance through verifier filtering, but in the *long run* the estimator converges to the verifier's knowledge center. This explains both the promise and the risk of such methods: a high-quality verifier can inject reliable external knowledge, while a biased verifier inevitably steers the model away from the truth. Viewed through the lens of *information elicitation*, our framework formalizes how external signals are incorporated recursively into training and why the outcome reflects the verifier's information.

Meanwhile, we also acknowledge the limitations of our results.Primarily, our analytical testbed relies on a well-specified parametric setting (linear regression) that assumes the existence of a global, ground-truth optimal parameter $\theta^*$. This idealized assumption mathematically abstracts the complex, often competing attributes of a "good" generative model—such as sample diversity and generation quality—into a singular distance metric. In practice, while models like LLMs are parametric, they are approximating a true data-generating process that is unknown and highly likely non-parametric. For complex domains like natural language, a singular "true model" may not even exist; therefore, defining or evaluating an optimal model remains an open challenge. While our empirical extensions to VAEs and LLMs validate the theory qualitatively, formal generalization to richer models such as exponential families or simple neural network architectures are interesting future directions. Other future venues include developing sharper bounds for nonlinear models, exploring effectiveness of alternative synthetic design strategies beyond block orthogonalization, and studying verifier dynamics in large language models (LLMs) and vision models.

**Acknowledgment.** This work is supported in part by the AI2050 program at Schmidt Sciences (Grant G-24-66104). Bingji Yi conducted this research while visiting UChicago CS. We also thank Cong Ma from UChicago, Hongning Wang and Bo Li from Tsinghua, Pinyan Lu and Gavin Tang from Shanghai University of Finance and Economics for helpful discussions and suggestions.

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

APPENDIX OVERVIEW

This appendix contains: Appendix A (1-D Gaussian toolkit), Appendix B (comparison with reward maximization frameworks), Appendix C (reduction and full proof for linear regression), Appendix D (additional details on VAE experiments), Appendix E (additional simulations and experiments), Appendix F (use of large language models)

## A   ONE-DIMENSIONAL GAUSSIAN TOOLKIT

In this section, we provide a toolkit for analyzing the one-dimensional Gaussian mean estimation problem with verifier-filtered synthetic data. This toolkit serves as the foundation for our analysis of the linear regression models. We will establish several key lemmas and theorems that characterize the MSE of the mean estimator under the one-dimensional Gaussian model. These results will be instrumental in proving Theorem 3.1 and Theorem 4.1 in Appendix C.

### A.1   SETUP AND NOTATIONS

We consider the one-dimensional mean estimation problem where the real data $X_1^0, \ldots, X_{n_0}^0$ are independently and identically distributed (i.i.d.) from a Gaussian distribution:

$$X_1^0, \ldots, X_{n_0}^0 \overset{\text{i.i.d.}}{\sim} \mathcal{N}(\mu, \sigma^2),$$

with known variance $\sigma^2$.

In our setting, a verifier exists and encodes external knowledge that the true mean lies in an interval $[a, b]$ (i.e. $\mu \in [a, b]$). Therefore, $\bar{X}^0 = \frac{X_1 + \cdots + X_{n_0}}{n_0}$ is the empirical mean of real data, which minimizes MSE if *no extra* information is supplied. We are interested in whether data verification could effectively inject new information and improve over $\bar{X}^0$. Consider the following synthetic data generation and filtering procedure:

- Generate $n_1$ synthetic data $X_1^1, \ldots, X_{n_1}^1 \overset{\text{i.i.d.}}{\sim} \mathcal{N}(\bar{X}^0, \sigma^2)$.

- Retain $X_i^0 \in [a, b]$ as $X_1'^1, \ldots, X_{n_1'}'^1$, and estimate $\mu$ using $\bar{X}^1 = \frac{1}{n_1'} \sum_{i=1}^{n_1'} X_i'^1$.

We will compare the estimator $\bar{X}^1$ with $\bar{X}^0$ and formally characterize when data verification enhances or degrades model performance - i.e., when $\mathbb{E}(\bar{X}^1 - \mu)^2 < \mathbb{E}(\bar{X}^0 - \mu)^2$ or not. Our key finding is that $\bar{X}^1$ introduces the core bias-variance trade-off that underpins model improvement or degradation. We will characterize the MSE of $\bar{X}^1$ which reveals how key quantities such as the real and synthetic sample size, the verifier's bias and variance will decide performance of the filtering strategy. These insights provide intuition for extending verifier-guided re-training to more complex settings.

We first review some notation and key results for the truncated normal distribution, which will be used in the subsequent sections. Consider a one-dimensional normal distribution $X \sim \mathcal{N}(\mu, \sigma^2)$ and let $X'$ be its truncated version restricted to the interval $[a, b]$. The distribution of $X'$ is the called the *truncated normal distribution*, denoted as $X' \sim \mathcal{N}(x|\mu, \sigma^2) \cdot \mathbb{1}_{\{a < x < b\}}$. The mean and variance of the truncated normal distribution $X'$ are given analytically:

$$\mathbb{E}[X'|\mu] = \mu - \sigma \frac{\phi(\frac{b-\mu}{\sigma}) - \phi(\frac{a-\mu}{\sigma})}{\Phi(\frac{b-\mu}{\sigma}) - \Phi(\frac{a-\mu}{\sigma})} := \mu + \sigma m_1\left(\frac{a-\mu}{\sigma}, \frac{b-\mu}{\sigma}\right)$$

$$\text{Var}(X'|\mu) = \sigma^2 \left[ 1 - \frac{\frac{b-\mu}{\sigma}\phi(\frac{b-\mu}{\sigma}) - \frac{a-\mu}{\sigma}\phi(\frac{a-\mu}{\sigma})}{\Phi(\frac{b-\mu}{\sigma}) - \Phi(\frac{a-\mu}{\sigma})} - \left( \frac{\phi(\frac{b-\mu}{\sigma}) - \phi(\frac{a-\mu}{\sigma})}{\Phi(\frac{b-\mu}{\sigma}) - \Phi(\frac{a-\mu}{\sigma})} \right)^2 \right]$$

$$:= \sigma^2 m_2\left(\frac{a-\mu}{\sigma}, \frac{b-\mu}{\sigma}\right) \tag{6}$$

where $\phi(x)$ and $\Phi(x)$ denote the standard normal density and cumulative distribution functions, respectively. Standardizing $X$ via $Z := \frac{X-\mu}{\sigma}$ and setting

$$\alpha = \frac{a-\mu}{\sigma}, \quad \beta = \frac{b-\mu}{\sigma}, \tag{7}$$

the expression in equation 6 become:

$$\mathbb{E}[Z'] = m_1(\alpha, \beta)$$
$$\mathrm{Var}(Z') = m_2(\alpha, \beta) \tag{8}$$

where $Z' \sim \mathcal{N}(x|0,1) \cdot \mathbb{1}_{\{\alpha < x < \beta\}}$ is the standardized truncated normal distribution. For convenience, we write $\mathcal{N}_{trunc}(\alpha, \beta) := \mathcal{N}(x|0,1) \cdot \mathbb{1}_{\{\alpha < x < \beta\}}$. Thus, $m_1$ and $m_2$ correspond to the first and second central moments of the standardized truncated normal distribution. In addition, we also define the third central moment of the standardized truncated normal distribution:

$$
\begin{aligned}
m_3(\alpha, \beta) &:= \mathbb{E}(Z' - \mathbb{E}Z')^3 \\
&= -\frac{(\beta^2 - 1)\phi(\beta) - (\alpha^2 - 1)\phi(\alpha)}{(\Phi(\beta) - \Phi(\alpha))} - \frac{3(\phi(\beta) - \phi(\alpha))(\beta\phi(\beta) - \alpha\phi(\alpha))}{(\Phi(\beta) - \Phi(\alpha))^2} \\
&\quad - \frac{2(\phi(\beta) - \phi(\alpha))^3}{(\Phi(\beta) - \Phi(\alpha))^3}.
\end{aligned}
\tag{9}
$$

In particular, $0 < m_2(\alpha, \beta) < 1$ for any $\alpha < \beta$ and $m_1(\alpha, \beta) = m_3(\alpha, \beta) = 0$ if $\alpha + \beta = 0$.

## A.2 CHARACTERIZATION OF $\mathbb{E}(\bar{X}^1 - \mu)^2$, BIAS-VARIANCE TRADE-OFF, AND MODEL IMPROVEMENT

**Theorem A.1.** *Assume that $n_1 > n_0 \geq 100$. Then there exists constant $K$, depending only on $\alpha$ and $\beta$, such that*

$$
\left| \frac{1}{\sigma^2} \mathbb{E}(\bar{X}^1 - \mu)^2 - \underbrace{\frac{m_2(\alpha, \beta)}{n_1}}_{\text{Synthetic Variance}} - \underbrace{\left( m_1^2(\alpha, \beta) + \frac{m_2^2(\alpha, \beta) + m_3(\alpha, \beta)m_1(\alpha, \beta)}{n_0} \right)}_{\text{Verification Bias+Variance}} \right|
$$
$$
< K \left( \frac{1}{n_1 n_0^{1/3}} + \frac{1}{n_0^{3/2}} \right)
\tag{10}
$$

*holds with probability at least $1 - \exp\left( -\frac{1}{2} n_0^{1/3} \right)$.*

*Proof of Theorem A.1.* It will be convenient to reparameterize the sample mean estimators by centering them around the true mean. Specifically, we define the residuals:

$$
\epsilon_1 := \frac{\bar{X}^0 - \mu}{\sigma}, \quad \epsilon_1 \sim \mathcal{N}(0, \frac{1}{n_0}).
\tag{11}
$$

Note that $\bar{X}^1$ is the mean of $n_1$ i.i.d. samples from the truncated normal distribution $\mathcal{N}(x|\bar{X}^0, \sigma^2) \cdot \mathbb{1}_{\{a < x < b\}}$. The MSE of $\bar{X}^1$ can be decomposed as follows:

$$
\begin{aligned}
\mathbb{E}[(\bar{X}^1 - \mu)^2] &= \mathbb{E}_{\bar{X}^0} \mathbb{E}_{\bar{X}^1 | \bar{X}^0} \left[ (\bar{X}^1 - \mu)^2 \right] \\
&= \mathbb{E}_{\bar{X}^0} \left[ \mathrm{Var}(\bar{X}^1 |, \bar{X}^0) + \left( \mathbb{E}[\bar{X}^1 | \bar{X}^0] - \mu \right)^2 \right] \\
&= \sigma^2 \mathbb{E}_{\bar{X}^0} \left[ \frac{m_2(\alpha - \epsilon_1, \beta - \epsilon_1)}{n_1} \right] + \mathbb{E}_{\bar{X}^0} \left[ (\bar{X}^0 - \mu - \sigma m_1(\alpha - \epsilon_1, \beta - \epsilon_1))^2 \right] \\
&= \frac{\sigma^2}{n_1} \mathbb{E}_{\epsilon_1} \left[ m_2(\alpha - \epsilon_1, \beta - \epsilon_1) \right] + \sigma^2 \mathbb{E}_{\epsilon_1} \left[ (m_1(\alpha - \epsilon_1, \beta - \epsilon_1) + \epsilon_1)^2 \right]
\end{aligned}
\tag{12}
$$

For the first term in 12, we consider the event $E_1 := \left\{ |\epsilon_1| < n_0^{-1/3} \right\}$, the function $m_2(\cdot, \cdot)$ is Lipschitz continuous in a neighborhood of $(\alpha, \beta)$, so we have

$$
|m_2(\alpha - \epsilon_1, \beta - \epsilon_1) - m_2(\alpha, \beta)| = |\epsilon_1| \cdot \left| m_2^{(1)}(\alpha - \xi, \beta - \xi) \right| < \frac{M_1}{n_0^{1/3}},
\tag{13}
$$

for some $\xi \in (0, \epsilon_1)$, where we define

$$M_1 := \sup_{|\xi| < \frac{1}{100^{\frac{1}{3}}}} \left| m_2^{(1)}(\alpha - \xi, \beta - \xi) \right|,$$

and $M_1$ is a constant independent of $n_0$ as long as $n_0 \geq 100$. Event $E_1$ hold with high probability:

$$\mathbb{P}\left( |\epsilon_1| < n_0^{-1/3} \right) > 1 - \frac{\exp\left( -\frac{n_0^{1/3}}{2} \right)}{\sqrt{\pi/2} \cdot n_0^{1/6}} > 1 - \frac{\exp\left( -\frac{n_0^{1/3}}{2} \right)}{\sqrt{\pi/2} \cdot 100^{1/6}} > 1 - \exp\left( -\frac{n_0^{1/3}}{2} \right).$$

Then we consider then second term in 12. The Taylor expansion of the function

$$m_1(\epsilon_1) := m_1(\alpha - \epsilon_1, \beta - \epsilon_1)$$

up to the third-order terms is:

$$m_1(\epsilon_1) = m_1(\alpha, \beta) - [1 - m_2(\alpha, \beta)]\,\epsilon_1 + \frac{1}{2}m_3(\alpha, \beta)\epsilon_1^2 + \frac{1}{6}m_1^{(3)}(\xi)\epsilon_1^3, \quad \text{for some } \xi \in (0, \epsilon_1), \tag{14}$$

where $m_1^{(3)}(\xi)$ denotes the third derivative of $m_1$ evaluated at some point between $0$ and $\epsilon_1$. Then we can get

$$\begin{aligned}
\mathbb{E}_{\epsilon_1}\left[ (m_1(\alpha - \epsilon_1, \beta - \epsilon_1) + \epsilon_1)^2 \right] &= \mathbb{E}\left( m_1(\alpha, \beta) + m_2(\alpha, \beta)\epsilon_1 + \frac{1}{2}m_3(\alpha, \beta)\epsilon_1^2 + \frac{1}{6}m_1^{(3)}(\xi)\epsilon_1^3 \right)^2 \\
&= m_1^2(\alpha, \beta) + \frac{m_2^2(\alpha, \beta) + m_1(\alpha, \beta)m_3(\alpha, \beta)}{n_0} + \frac{3m_3^2(\alpha, \beta)}{4n_0^2} \\
&\quad + \mathbb{E}\left( m_1(\alpha, \beta) + m_2(\alpha, \beta)\epsilon_1 + \frac{1}{2}m_3(\alpha, \beta)\epsilon_1^2 \right) \frac{m_1^{(3)}(\xi)}{3} \epsilon_1^3 \\
&\quad + \mathbb{E}\left( \frac{m_1^{(3)^2}(\xi)}{36} \epsilon_1^6 \right). 
\end{aligned} \tag{15}$$

First, using the fact that there exists constant $M$ that only depends on $\alpha$ and $\beta$, such that $|m_1^{(3)}(x)| < M$ for any $x$, we have:

$$\begin{aligned}
&\left| \mathbb{E}\left[ \left( m_1(\alpha, \beta) + m_2(\alpha, \beta)\epsilon_1 + \frac{1}{2}m_3(\alpha, \beta)\epsilon_1^2 \right) \frac{m_1^{(3)}(\xi)}{3} \epsilon_1^3 \right] \right| \\
&\leq \mathbb{E}\left[ \left( |m_1(\alpha, \beta)| + m_2(\alpha, \beta)|\epsilon_1| + \frac{1}{2}|m_3(\alpha, \beta)|\epsilon_1^2 \right) \cdot \frac{M}{3}|\epsilon_1|^3 \right] \\
&= \mathbb{E}\left[ \frac{M}{3}|m_1(\alpha, \beta)||\epsilon_1|^3 + \frac{M}{3}m_2(\alpha, \beta)|\epsilon_1|^4 + \frac{M}{6}|m_3(\alpha, \beta)||\epsilon_1|^5 \right] \\
&\leq \frac{K_1}{n_0^{3/2}}.
\end{aligned}$$

for some constant $K_1$ depending only on $\alpha$ and $\beta$.

Secondly, the last term in equation 15 is bounded by:

$$\mathbb{E}\left[ \frac{m_1^{(3)^2}(\xi)}{36} \epsilon_1^6 \right] \leq \frac{M^2}{36}\mathbb{E}[\epsilon_1^6] = \frac{5M^2\sigma^6}{12n_0^3} \leq \frac{K_2}{n_0^3},$$

for some constant $K_2$.

So the second term in 12 is bounded by

$$\left| \mathbb{E}_{\epsilon_1}\left[ (m_1(\alpha - \epsilon_1, \beta - \epsilon_1) + \epsilon_1)^2 \right] - m_1^2(\alpha, \beta) - \frac{m_2^2(\alpha, \beta) + m_1(\alpha, \beta)m_3(\alpha, \beta)}{n_0} \right| < \frac{K}{n_0^{3/2}} \tag{16}$$

for some constant $K$.

Combining 12, 13, and 16 completes the proof.

$\square$

### A.3 ITERATIVE RETRAINING AND LONG-TERM DYNAMICS IN ONE-DIMENSIONAL GAUSSIAN MEAN ESTIMATION

Now consider the verifier-guided synthetic retraining in the Gaussian mean estimation setting. The iterative retraining process can be described by the following algorithm.

---

**Algorithm 1** Iterative Verifier-Guided Retraining for Gaussian Mean Estimation

---

1: **Input:** Initial estimate $\bar{X}^0$ from real data
2: **for** $k = 0, 1, 2, \ldots$ **do**
3:     Draw $\xi_i \overset{\text{i.i.d.}}{\sim} \mathcal{N}(0, \sigma^2)$ and construct synthetic samples $X_i^k = \bar{X}^k + \xi_i$.
4:     Retain points with $a < X_i^k < b$, yielding $n_k$ verified samples $\{X_i'^k : i = 1, 2, \ldots n_k\}$.
5:     $\bar{X}^{k+1} \leftarrow \frac{1}{n_k} \sum_{i=1}^{n_k} X_i'^k$.
6: **end for**

---

Algorithm 1 defines a Markov process $\{\bar{X}^0, \bar{X}^1, \ldots \bar{X}^k, \ldots\}$, where the conditional distribution $p(\bar{X}^{k+1}|\bar{X}^k)$ is given by

$$p(\bar{X}^{k+1}|\bar{X}^k) : \bar{X}^{k+1} = \bar{X}^k + \sigma \frac{\sum_{i=1}^{n_k} \xi_i'^{k+1}}{n_k}, \qquad \xi_i'^{k+1} \text{ i.i.d } \sim \mathcal{N}_{trunc}(\frac{a - \bar{X}^k}{\sigma}, \frac{b - \bar{X}^k}{\sigma}) \quad (17)$$

The following theorem summarizes these findings:

**Theorem A.2.** *Let $\bar{X}^k$ be the Markov process determined by equation 17 with initial condition*

$$\bar{X}^0 \sim \mathcal{N}(0, \frac{\sigma^2}{n_0}),$$

*and assume $n_k$ is non-decreasing in $k$. Then the following statements hold:*

- *If $|a|, |b| < \infty$, there exists a constant $0 < \rho < 1$ such that,*

$$\mathbb{E}\left(\bar{X}^k - \frac{a+b}{2}\right)^2 \leq \rho^{2k} \mathbb{E}(\bar{X}^0 - \frac{a+b}{2})^2 + \sum_{j=0}^{k-1} \frac{\rho^{2(k-j)-1}}{n_j}.$$

  *Moreover, if $\lim_{k\to\infty} n_k = \infty$, $\lim_{k\to\infty} \mathbb{E}|\bar{X}^k - \frac{a+b}{2}|^2 = 0$.*

- *If $-\infty = a \leq b < \infty$, then $\liminf_{k\to\infty} \bar{X}^k = -\infty$. If $-\infty < a < b = \infty$, then $\limsup_{k\to\infty} \bar{X}^k = \infty$.*

*Proof of Theorem A.2.* Define

$$\epsilon_k = \frac{\bar{X}^k - \mu}{\sigma}, \tag{18}$$

which represents the standardized error of the estimator $\bar{X}^k$. It is easy to see that $\epsilon_k \in [\alpha, \beta] \Leftrightarrow \bar{X}^k \in [a, b]$, where $\alpha, \beta$ are defined in equation 7. Therefore, it suffices to consider the standardized process $\{\epsilon_k, k = 0, 1, 2, \ldots\}$. equation 17 can be standardized as:

$$\epsilon_{k+1} = \epsilon_k + \frac{\sum_{i=1}^{n_k} \xi_i'^{k+1}}{n_k}, \qquad \xi_i'^{k+1} \sim \mathcal{N}_{\text{trunc}}(\alpha - \epsilon_k, \beta - \epsilon_k), \tag{19}$$

For convenience, we shift the noise terms $\xi_i'^{k+1}$ in equation 19 to have mean zero. Therefore, we introduce

$$T_{\alpha,\beta}(x) := x + \mathbb{E}[Z \mid \alpha - x \leq Z \leq \beta - x], \qquad v_{\alpha,\beta}(x) := \text{Var}(Z \mid \alpha - x \leq Z \leq \beta - x). \tag{20}$$

where $Z \sim \mathcal{N}(0, 1)$.

Therefore, equation 19 can be rewritten as

$$\epsilon_{k+1} = T_{\alpha,\beta}(\epsilon_k) + \eta_{k+1} \tag{21}$$

where $\eta_{k+1} = \frac{1}{n_k} \sum_{i=1}^{n_k} \left( \xi_i'^{k+1} - \mathbb{E}\xi_i'^{k+1} \right)$ is the average of independent mean zero noise in equation 19. In particular, we have

$$\mathbb{E}[\eta_{k+1} \mid \mathcal{F}_k] = 0, \qquad \text{Var}(\eta_{k+1} \mid \mathcal{F}_k) = \frac{v_{\alpha,\beta}(\epsilon_k)}{n_k}.$$

where $\mathcal{F}_k := \sigma(\epsilon_0, \eta_1, \ldots, \eta_k)$ and $n_k$ is the (post-filtering) batch size at round $k$.

It is easy to see that

$$\begin{aligned}
T_{\alpha,\beta}(x) &= x + m_1(\alpha - x, \beta - x), \\
v_{\alpha,\beta}(x) &= m_2(\alpha - x, \beta - x), \\
T'_{\alpha,\beta}(x) &= v_{\alpha,\beta}(x).
\end{aligned}$$

We first consider $|a|, |b| < \infty$. In this case, we first show that the derterministic part $T_{\alpha,\beta}(x)$ in equation 21 is a global contraction. Since $-\infty < \alpha < \beta < \infty$, we have

$$\sup_{x \in \mathbb{R}} T'_{\alpha,\beta}(x) = \sup_{x \in \mathbb{R}} \text{Var}\left(Z \mid \alpha - x \leq Z \leq \beta - x\right) = \text{Var}\left(Z \mid |Z| < |\frac{\alpha + \beta}{2}|\right) := \rho < 1.$$

Therefore, $T_{\alpha,\beta}(x)$ is a global contraction. By the contractive mapping theorem that $T_{\alpha,\beta}(x)$ has a unique fixed point $x^*$, which solves $x^* = T_{\alpha,\beta}(x^*)$. It is easy to see that

$$x^* = T_{\alpha,\beta}(x^*) \implies x^* = x^* + \mathbb{E}(Z | \alpha - x^* \leq Z \leq \beta - x^*) \implies x^* = \frac{\alpha + \beta}{2}. \tag{22}$$

By the mean-value theorem,

$$\left| T_{\alpha,\beta}(\epsilon_k) - \frac{\alpha + \beta}{2} \right| \leq \rho \left| \epsilon_k - \frac{\alpha + \beta}{2} \right|.$$

Let $V_k := (\epsilon_k - \frac{\alpha+\beta}{2})^2$, we have

$$\mathbb{E}[V_{k+1} \mid \epsilon_k] = (T_{\alpha,\beta}(\epsilon_k) - \frac{\alpha + \beta}{2})^2 + \frac{v_{\alpha,\beta}(\epsilon_k)}{n_k} \leq \rho^2(\epsilon_k - \frac{\alpha + \beta}{2})^2 + \frac{\rho}{n_k}.$$

Taking expectations yields

$$\mathbb{E}V_{k+1} \leq \rho^2 \mathbb{E}V_k + \frac{\rho}{n_k}. \tag{23}$$

Unrolling equation 23,

$$\mathbb{E}V_k \leq \rho^{2k}\mathbb{E}V_0 + \rho \sum_{j=0}^{k-1} \frac{\rho^{2(k-1-j)}}{n_j}. \tag{24}$$

It is easy to see that

$$\mathbb{E}V_k \leq \rho^{2k}\mathbb{E}V_0 + \rho \sum_{j=0}^{k-1} \frac{\rho^{2(k-1-j)}}{n_0} < \rho^{2k}\mathbb{E}V_0 + \frac{\rho}{n_0(1 - \rho^2)}.$$

Therefore, by the Cauchy-Schwarz inequality, $\lim_{k\to\infty} \mathbb{E}\epsilon_k^2 < \infty$ easily follows. Moreover, when $n_k \to \infty$, let $g_i := \rho^{2i}$ and $a_j := 1/n_j \to 0$. A standard $\ell^1$-convolution argument shows $(g * a)_k := \sum_{j=0}^{k-1} g_{k-1-j} a_j = \sum_{j=0}^{k-1} \frac{\rho^{2(k-1-j)}}{n_j} \to 0$. Therefore $\lim_{k\to\infty} \mathbb{E}V_k = \lim_{k\to\infty} \mathbb{E}(\epsilon_k - \frac{\alpha+\beta}{2})^2 = 0$.

Now we consider the case $-\infty = a < b < \infty$ (equivalently $-\infty = \alpha < \beta < \infty$). We will show that $\liminf_{k\to\infty} \epsilon_k = -\infty$ a.s..

Let $t_k := \beta - \epsilon_k$ and the recursion equation 21 can be rewritten for $t_k$:

$$t_{k+1} = t_k + \lambda(t_k) - \eta_{k+1},$$

where $\lambda(t_k) = -\mathbb{E}(Z | Z < \beta - \epsilon_k) = \mathbb{E}[Z \mid Z \geq -t_k]$.

Consider the hitting time $\tau_M := \inf\{k : t_k \geq M\}$ for any $M > 0$. Fix $M > 0$ and define

$$m(M) := \min_{t \leq M} \lambda(t) = \mathbb{E}[Z \mid Z \geq -M] > 0,$$

which is strictly positive the fact that $\lambda(t) > 0$ and $\lambda(t)$ is a decreasing function. On the event $\{\tau_M > K\}$ we have $t_j < M$ for $j = 0, \ldots, K-1$, hence $\lambda(t_j) \geq m(M)$. Summing the recursion yields

$$t_K = t_0 + \sum_{j=0}^{K-1} \lambda(t_j) - \sum_{j=0}^{K-1} \eta_{j+1} \geq t_0 + K\, m(M) - S_K,$$

where $S_K := \sum_{j=0}^{K-1} \eta_{j+1}$ and $t_0 = \beta - \epsilon_0$ is $\mathcal{F}_0$-measurable (hence random). Therefore,

$$\{\tau_M > K\} \subseteq \left\{ S_K \geq t_0 + K\, m(M) - M \right\}. \tag{25}$$

Define the (random) burn-in index

$$K_0 := \left\lceil \frac{2(M - t_0)}{m(M)} \right\rceil.$$

Then for all $K \geq K_0$,

$$t_0 + K\, m(M) - M \geq \frac{m(M)}{2} K,$$

and equation 25 gives, conditionally on $\mathcal{F}_0$,

$$\{\tau_M > K\} \subseteq \left\{ S_K \geq \frac{m(M)}{2} K \right\}, \qquad \text{for all } K \geq K_0. \tag{26}$$

Next, we will show that $S_K$ is a sub-exponential random variable in event $\{\tau_M > K\}$. Since $S_K = \sum_{j=0}^{K-1} \eta_{j+1} = \sum_{j=0}^{K-1} \frac{1}{n_j} \sum_{i=1}^{n_j} \left( \xi_i'^{j+1} - \mathbb{E}\xi_i'^{j+1} \right)$, we will first show that $\xi_i'^{j+1} - \mathbb{E}\xi_i'^{j+1}$ is sub-exponential.

Since $\xi_i'^{j+1} \sim \mathcal{N}_{\text{trunc}}(-\infty, \beta - \epsilon_j) = \mathcal{N}_{\text{trunc}}(-\infty, t_j)$, on the event $\{\tau_M > K\}$ we have

$$\xi_i'^{j+1} - \mathbb{E}\xi_i'^{j+1} < t_j - \mathbb{E}[Z \mid Z < t_j] \leq M - \mathbb{E}[Z \mid Z < M] := b(M) < \infty.$$

The above inequality follows from the fact that $t - \mathbb{E}[Z \mid Z < t]$ is an increasing function of $t$ and $t_j < M$ for $j = 0, \ldots, K-1$ on the event $\{\tau_M > K\}$. In addition, $\text{Var}(\xi_i'^{j+1}) = \text{Var}(Z|Z < t_j) \leq 1$. Therefore, $\xi_i'^{j+1} - \mathbb{E}\xi_i'^{j+1}$ is mean zero, bounded above by $b(M)$ with $\text{Var}\left( \xi_i'^{j+1} - \mathbb{E}\xi_i'^{j+1} \right) < 1$. By Bennet/Bernstein MGF inequality, we have

$$\log \mathbb{E} e^{\lambda(\xi_i'^{j+1} - \mathbb{E}\xi_i'^{j+1})} \leq \frac{\lambda^2}{2(1 - b(M)\lambda/3)},$$

for $0 < \lambda < \frac{3}{b(M)}$. This shows that $\xi_i'^{j+1} - \mathbb{E}\xi_i'^{j+1}$ is sub-exponential with parameters $SE(1, 2b(M)/3)$. By standard properties of sub-exponential random variables, $\eta_{j+1} = \frac{1}{n_j} \sum_{i=1}^{n_j} \left( \xi_i'^{j+1} - \mathbb{E}\xi_i'^{j+1} \right)$ is $SE(1/n_j, 2b(M)/(3n_j))$ and $S_K = \sum_{j=0}^{K-1} \eta_{j+1}$ is $SE(\sum_{j=0}^{K-1} 1/n_j, 2b(M)/(3n_1))$ since $n_j$ is non-decreasing. Therefore, for any $t > 0$ we have tail bound

$$\mathbb{P}\left( S_K \geq t \right) \leq \exp\left( -\frac{1}{2} \min\{ \frac{t^2}{\sum_{j=0}^{K-1} 1/n_j}, \frac{n_1 t}{2b(M)} \} \right) \leq \exp\left( -\frac{1}{2} \min\{ \frac{n_1 t^2}{K}, \frac{n_1 t}{2b(M)} \} \right). \tag{27}$$

Use the tail bound equation 27 in equation 26, we have

$$\mathbb{P}\left(\tau_M > K \mid \mathcal{F}_0\right) \leq \mathbb{P}\left(S_K \geq \frac{m(M)}{2}K\right) \leq \exp\left(-c(M)n_1 K\right) \tag{28}$$

for all $K \geq K_0$ with $c(M) = \min\left\{\frac{m(M)^2}{8}, \frac{n(M)}{8b(M)}\right\}$.

$$\mathbb{P}\left(\tau_M > K\right) = \mathbb{E}\left[\mathbb{P}\left(\tau_M > K \mid \mathcal{F}_0\right)\right]$$
$$\leq \mathbb{E}\left[\exp\left(-c(M)n_1 K\right)\mathbb{1}_{\{K > K_0\}}\right] + \mathbb{P}\left(K \leq K_0\right) \tag{29}$$

Let $K \to \infty$ in equation 29, we get $\mathbb{P}(\tau_M < \infty) = 1$. Since $M$ is arbitrary, this implies $\liminf_{k\to\infty} \epsilon_k = -\infty$ a.s..

The case $-\infty < a < b = \infty$ can be proved in the same way, therefore is omitted.

$\square$

## B  COMPARISON WITH REWARD MAXIMIZATION FRAMEWORKS

While sharing the conceptual similarity of evaluating generated data via an external feedback mechanism, our modeling approach substantially differs from reward maximization frameworks in various aspects, including preference matching (Ferbach et al., 2024; Wei & Zhang) and recent Reinforcement Learning with Verified Rewards (RLVR)(Guo et al., 2024; Yu et al., 2025). Theoretically, these methods frame the problem as policy optimization, where the objective is to maximize a provided reward signal. While highly effective for alignment, the definition of a "good model" is tied to the specific reward formulation, which may not correspond to recovering the true data-generating distribution. Practically, reward optimization relies on assigning scalar rewards or pairwise comparison signals (Ouyang et al., 2022), which are often difficult and noisy to define. For instance, evaluating image quality or open-ended language generation with a single numerical reward is inherently subjective. While recent methods like RLVR avoid this issue by restricting themselves to domains with clearly verifiable rewards (Guo et al., 2025; Wu & Choi; Yu et al., 2025), many important training settings lack such reliable reward functions. In contrast, we study the widely used "generate-verify-retrain" paradigm, which utilizes binary accept/reject filtering. Theoretically, our framework defines a "good model" at the parameter level, explicitly modeling the relationship between the verifier's filtering rule and the ground truth. By formalizing this link, we can directly analyze model performance during iterative retraining, even for an imperfect or biased verifier. Practically, this filtering mechanism is less noisy, highly stable, and serves as a core scalable primitive in modern LLM pipelines like DeepSeek-Coder (Guo et al., 2025).

## C  PROOFS OF ALL THEOREMS IN SECTION 2

---

**Scheme 2** Iterative Retraining with Verified Synthetic Data [5]

1: **Input:** Initial estimator $\hat{\theta}^0$ from $n_0$ real samples
2: **for** $k = 0, 1, 2, \ldots$ **do**
3:     **Generate:** Synthetic covariates $X^{k+1}$ are constructed (details below), and responses are generated as:
$$Y^{k+1} = X^{k+1}\hat{\theta}^k + \xi^{k+1}, \quad \xi^{k+1} \sim \mathcal{N}(0, \sigma^2 I)$$
4:     **Verify:** Each synthetic sample $(x_i^{k+1}, y_i^{k+1})$ of $(X^{k+1}, Y^{k+1})$ is filtered by the verifier Condition equation 1, retaining only the verified subset $(X^{k+1\prime}, Y^{k+1\prime})$.
5:     **Retrain:** A new OLS estimator is computed using only the verified data:
$$\hat{\theta}^{k+1} = (X^{k+1\prime\top} X^{k+1\prime})^{-1} X^{k+1\prime\top} Y^{k+1\prime}$$
6: **end for**

---

Given the orthogonality of $\{v_j\}$ in the block design equation 2, the OLS estimator decomposes into a set of one-dimensional problems, each estimating the coordinate of $\theta$ along direction $v_j$. In particular, choosing $\{v_j\}$ as the right singular vectors of the real data matrix $X^0$ yields the cleanest interpretation, making explicit how verifier bias, synthetic sample size, and noise variance interact. Accordingly, the retraining procedure can be formalized as follows:

---

[4]Since only the most recent round of synthetic data is retained for training, this scheme is sometimes referred to as a *discard workflow* in the literature (e.g., (Dey & Donoho, 2024)).

---

**Algorithm 3** Iterative Verifier-Guided Retraining in Linear Regression

1: **Input:** Real data $(X^0, Y^0)$
2: Compute initial estimator $\hat{\theta}^0 = (X^{0\top} X^0)^{-1} X^{0\top} Y^0$
3: Let $X^0 = U\Sigma V^\top$ be the SVD of $X^0$, with right singular vectors $V = (v_1, \ldots, v_p)$
4: **for** $k = 0, 1, 2, \ldots$ **do**
5:      **for** $j = 1, \ldots, p$ **do**
6:          Construct synthetic design matrix $X^{k+1,j}$ with all rows equal to $v_j^\top$
7:          Generate synthetic responses $Y^{k+1,j} = X^{k+1,j}\hat{\theta}^k + \xi^{k+1,j}$, where $\xi^{k+1,j} \sim \mathcal{N}(0, \sigma^2 I)$
8:          Apply verifier to each $(x_i^{k+1,j}, y_i^{k+1,j})$ and retain valid samples satisfying

$$|y_i^{k+1,j} - (x_i^{k+1,j})^\top \theta_c| \le r\|x_i^{k+1,j}\| + \sigma_c, \tag{30}$$

9:          yielding $n_k$ verified samples $(x_i'^{k+1,j}, y_i'^{k+1,j})$.
10:          Compute one-dimensional estimator

$$\hat{\theta}^{k+1,proj,j} = \bar{y}'^{k+1,j} \tag{31}$$

11:      **end for**
12:      Update overall estimator:

$$\hat{\theta}^{k+1} = \sum_{j=1}^{p} v_j \hat{\theta}^{k+1,proj,j} \tag{32}$$

13: **end for**

---

*Proof of Theorem 3.1.* We consider the one dimensional projection estimator of $\hat{\theta}^{1,proj,j}$ defined in equation 31. The filter condition equation 30 is equivalent to:

$$|\sigma\xi_i^{1,j} + v_j^\top(\hat{\theta}^0 - \theta_c)| \le r + \sigma_c$$
$$\Longleftrightarrow y_i^{1,j} = \sigma\xi_i^{1,j} + v_j^\top\hat{\theta}^0 \in \left(-r - \frac{\sigma_c}{\sigma} + v_j^\top\theta_c, r + \frac{\sigma_c}{\sigma} + v_j^\top\theta_c\right). \tag{33}$$

Note that $\hat{\theta}^0 \sim \mathcal{N}(\theta^\star, (X^{0\top}X^0)^{-1}\sigma^2)$ and $v_j$ is the $j$-th right singular vector of $X^0$, therefore $v_j^\top\hat{\theta}^0 \sim \mathcal{N}(v_j^\top\theta^\star, \sigma^2\mu_j^{-2})$. Therefore, $\hat{\theta}^{1,proj,j} = \bar{y}'^{1,j}$ correspond to the verifier-filtered mean estimator of a one-dimensional Gaussian mean estimation problem with true mean $v_j^\top\theta$, variance $\sigma^2\mu_j^{-2}$ and filtering interval $\left(-r - \frac{\sigma_c}{\sigma} + v_j^\top\theta_c, r + \frac{\sigma_c}{\sigma} + v_j^\top\theta_c\right)$. Let

$$\alpha_j := \frac{-r - \sigma_c + v_j^\top(\theta_c - \theta^\star)}{\sigma},$$
$$\beta_j := \frac{r + \sigma_c + v_j^\top(\theta_c - \theta^\star)}{\sigma}. \tag{34}$$

Under the assumption $\mu_j = \omega(\sqrt{n_0})$, there exists a constant $L > 0$, such that $\mu_j^2 > Ln_0$ for all $j = 1, \ldots, p$. Therefore, by Theorem A.1, there exists constant $K_j$ depending only on $\alpha_j, \beta_j$ such that if $n_1 > n_0 \ge 100$,

$$\left| \frac{1}{\sigma^2}\mathbb{E}(\hat{\theta}^{1,proj,j} - v_j^\top\theta^\star)^2 - \frac{m_2(\alpha_j, \beta_j)}{n_1} - \left( m_1^2(\alpha_j, \beta_j) + \frac{m_2^2(\alpha_j, \beta_j) + m_3(\alpha_j, \beta_j)m_1(\alpha_j, \beta_j)}{\mu_j^2} \right) \right|$$
$$< K_j\left(\frac{1}{n_1 n_0^{1/3}} + \frac{1}{n_0^{3/2}}\right) \tag{35}$$

will hold with probability at least $1 - \exp(-Ln_0^{1/3})$. $m_1, m_2, m_3$ are defined in equation 8 and equation 9. By equation 32, we have $\hat{\theta}^{1,proj,j} = v_j^\top\hat{\theta}^1$. In addition, since $V = (v_1, v_2, \ldots, v_p)$ is an orthonormal matrice, we have

$$\sum_{j=1}^{p}\mathbb{E}(\hat{\theta}^{1,proj,j} - v_j^\top\theta^\star)^2 = \sum_{j=1}^{p}\mathbb{E}(v_j^\top\hat{\theta}^1 - v_j^\top\theta^\star)^2 = \mathbb{E}\|V^\top(\hat{\theta}^1 - \theta^\star)\|^2 = \mathbb{E}\|\hat{\theta}^1 - \theta^\star\|^2. \tag{36}$$

Therefore, by summing over $j$ on both sides of equation 35 and using simple union bound, we have

$$\left| \frac{1}{\sigma^2}\mathbb{E}\|\hat{\theta}^1 - \theta^\star\|^2 - \sum_{j=1}^{p}\left( \underbrace{\frac{m_{2,j}}{n_1}}_{\text{Synthetic Variance}} + \underbrace{m_{1,j}^2 + \frac{m_{1,j}m_{3,j} + m_{2,j}^2}{\mu_j^2}}_{\text{Verification Error}} \right) \right| < K\left( \frac{1}{n_1 n_0^{1/3}} + \frac{1}{n_0^{3/2}} \right)$$

(37)

with $K = \max_j K_j$ and

$$m_{1,j} := m_1(\alpha_j, \beta_j),$$
$$m_{2,j} := m_2(\alpha_j, \beta_j),$$
$$m_{3,j} := m_3(\alpha_j, \beta_j).$$

$\square$

*Proof of Theorem 4.1.* We consider the transition dynamics of $\hat{\theta}^k$ in Algorithm 3. Since we designed $X^{k+1,j}$ to be the rank one matrix correspond to singular vector $v_j$, therefore equation 31 can be rewritten as

$$\hat{\theta}^{k+1,proj,j} = v_j^\top \hat{\theta}^k + \frac{\sigma}{n_k}\sum_{i=1}^{n_k}\xi_i'^{k+1,j}$$

(38)

where $\xi_i'^{k+1,j}$ is the truncated noise term after verification. By equation 30, we have

$$\xi_i'^{k+1,j} \text{ i.i.d} \sim \mathcal{N}_{trunc}\left( -\frac{r}{\sigma} - \frac{\sigma_c}{\sigma} - v_j^\top\frac{\hat{\theta}^k - \theta_c}{\sigma}, \frac{r}{\sigma} + \frac{\sigma_c}{\sigma} - v_j^\top\frac{\hat{\theta}^k - \theta_c}{\sigma} \right).$$

(39)

We consider the rotated standardized estimator

$$\epsilon_j^k := v_j^\top\frac{\hat{\theta}^k - \theta_c}{\sigma} \quad \text{equivalently} \quad \epsilon^k := V^\top\frac{\hat{\theta}^k - \theta_c}{\sigma}.$$

Since $\hat{\theta}^{k+1,proj,j} = v_j^\top\hat{\theta}^{k+1}$ by equation 32, equation 38 can be standardized as

$$\epsilon_j^{k+1} = \epsilon_j^k + \frac{\sum_{i=1}^{n_k}\xi_i'^{k+1,j}}{n_k}, \qquad \xi_i'^{k+1,j} \text{ i.i.d} \sim \mathcal{N}_{trunc}\left( -\beta - \epsilon_j^k, \beta - \epsilon_j^k \right)$$

(40)

where $\beta = \frac{r}{\sigma} + \frac{\sigma_c}{\sigma}$. We note that equation 40 is exactly the same dynamics we consider in the proof of Theorem A.2 with $\beta = -\alpha < \infty$. In other words, the evolution of the iterative estimator $\epsilon^k$ is diagonal and each cordinates follows the same dynamics as the one dimensional gaussian iterative mean estimator. From Theorem A.2, we known that there exists a constant $\rho < 1$ such that

$$\mathbb{E}\|\epsilon_j^k\|^2 \le \rho^{2k}\mathbb{E}\|\epsilon_j^0\|^2 + \sum_{j=0}^{k-1}\frac{\rho^{2(k-j)-1}}{n_j}, \qquad j = 1, 2, \ldots, p.$$

This implies that

$$\mathbb{E}\|\hat{\theta}^k - \theta_c\|^2 \le \rho^{2k}\mathbb{E}\|\hat{\theta}^0 - \theta_c\|^2 + p\sigma^2\sum_{j=0}^{k-1}\frac{\rho^{2(k-j)-1}}{n_j}.$$

$\square$

# D  ADDITIONAL DETAILS ON CVAE EXPERIMENTS

**Data preprocessing.**  We use MNIST ($28 \times 28$ grayscale) and normalize pixel intensities to $[0, 1]$. Class labels are represented as one-hot vectors $y \in \{0, 1\}^K$ ($K{=}10$).

**Experiment Details.**  We use a convolutional CVAE model consisting of an Encoder with two convolutional layers ($1{\to}32$ and $32{\to}64$ channels, $4 \times 4$ kernels, stride 2, with GELU activations), followed by a linear projection that outputs the mean and log-variance of a $d_z = 20$-dimensional Gaussian latent space. The Decoder mirrors this structure: a linear layer maps the latent code to a $64 \times 7 \times 7$ tensor, which is upsampled by two transposed convolutional layers ($64{\to}32$ and $32{\to}1$ channels, $4 \times 4$ kernels, stride 2, with GELU activations) to reconstruct $28 \times 28$ images. We train the CVAE with the standard objective, i.e., binary cross-entropy reconstruction loss plus KL divergence regularization.

**Discriminator for filtering.**  We additionally train a discriminator $D$ to distinguish real from synthetic samples. $D$ is implemented as a multi-layer perceptron: five fully connected layers with hidden sizes 512, 256, 128, and 64, each followed by a LeakyReLU activation, and a final linear layer mapping to a single logit. The output is passed through a sigmoid to yield the probability of the input being real. The discriminator is trained with binary cross-entropy, labeling real MNIST digits as positive and CVAE-generated digits as negative.

**Synthetic generation and filtering.**  After each training round, we generate conditioned samples by drawing $z \sim \mathcal{N}(0, I)$, choosing labels $y$ (uniform over classes unless specified), and decoding $\tilde{x} = g_\theta(z, y)$. To control sample quality, we score each $(\tilde{x}, y)$ with the discriminator $D(\tilde{x}, y)$. For each class, we retain only the top $10\%$ of generated samples with the highest discriminator scores. These filtered synthetic samples are then combined with the real dataset to form the training data for the next round.

**Supplementary Results on ELBO**  We also evaluate generative performance using the test negative ELBO, a standard likelihood-based loss metric for VAEs. To prevent overfitting the discriminator (i.e., the verifier) and ensure stability during the retraining cycles, we incorporate standard regularization techniques, specifically applying a dropout rate of 0.1 and label smoothing with a parameter of 0.05 when training the discriminator. To investigate the effect of synthetic data size $n_k$, we employ three linearly increasing sample size schedules. Starting with an initial CVAE trained on only 500 samples, we scale up the retraining size by adding 5K, 30K, or 50K synthetic samples per iteration, respectively. The models are retrained for 50 iterations until the test negative ELBO stabilizes.

Figure 6 reports the test negative ELBO over these 50 rounds. Consistent with our bias–variance analysis, we observe a clear improvement (a decrease in loss) in the early stages (up to roughly iterations 10–15). Furthermore, the trajectories reveal a critical dynamic: while larger synthetic size schedules significantly accelerate this early convergence, all three schedules ultimately plateau and converge to a similar negative ELBO value by iteration 50. This observation validates our theoretical framework: drawing more synthetic samples expedites the initial variance reduction phase, but the asymptotic performance limit is dictated by the verifier, not the volume of synthetic data. After the initial variance-reduction gains, the negative ELBO eventually reverses its trend and increases (deteriorates) as the model inevitably converges toward the verifier's knowledge center.

As discussed in the main text regarding our verifier's limitations, this knowledge center is demonstrably biased. For reference, across all three size schedules, the final retrained CVAEs at iteration 50 converge to a test negative ELBO of approximately 111. In contrast, a baseline CVAE trained on the entire 60K real image dataset attains a test negative ELBO of 92.12 (lower is better). Because our verifier emphasizes perceptual quality over likelihood-based reconstruction, the negative ELBO proves harder to improve than FID. As a result, even as the negative ELBO stagnates or worsens in later iterations, our retrained models continue to improve FID, achieving sharper, cleaner digits. We believe that deploying stronger verifiers with, e.g., diversity preservation capabilities could enable iterative retraining to further improve the negative ELBO.

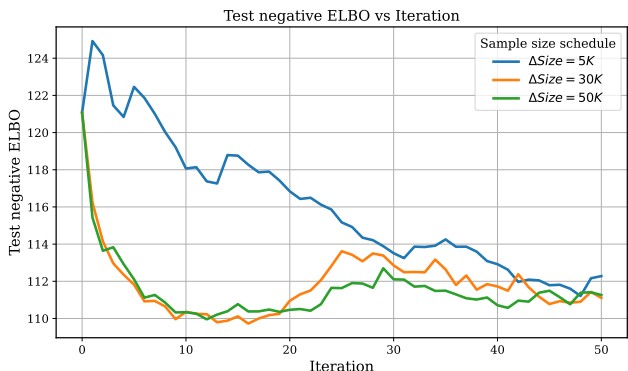

Figure 6: Test negative ELBO across retraining iterations.

# E    ADDITIONAL EXPERIMENTAL RESULTS

## E.1    RANDOM SYNTHETIC DATA IN LINEAR REGRESSION

In the main text, the synthetic covariates were aligned with a fixed orthonormal basis to simplify analysis and make the retraining dynamics easier to interpret. To show that the observed behavior is not tied to this structured design, we repeat the same iterative retraining experiment using fully random synthetic covariates sampled i.i.d. from a standard Gaussian distribution.

Figure 7 presents the results, corresponding directly to the lower two panels in Figure 3 of the main text, but under the random-design setting. The qualitative behavior remains the same: with a well-specified verifier, retraining contracts toward the verifier's knowledge center and avoids collapse, whereas unfiltered retraining diverges. This confirms that the verifier-induced stability and improvement patterns hold beyond the orthonormal-design assumption.

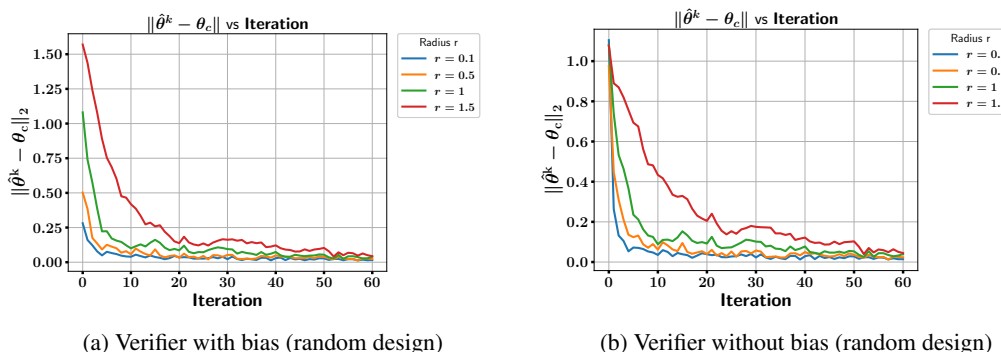

(a) Verifier with bias (random design)    (b) Verifier without bias (random design)

Figure 7: Iterative synthetic retraining under random synthetic covariates, corresponding to the structured-design results in Figure 3.

## E.2    DIFFERENT VERIFIER SHAPES

We further analyze how different geometric choices of the verifier region affect the acceptance rule and the resulting retraining dynamics. For any region $\mathcal{R}_\theta$ around a center $\theta_c$, a synthetic point $(x, y)$ is accepted whenever there exists a parameter perturbation $\Delta$ in the region that can explain $y$, i.e.

$$y = x^\top(\theta_c + \Delta) + \xi, \qquad \Delta \in \mathcal{R}_\theta.$$

This leads to the general acceptance requirement

$$|y - x^\top\theta_c| \leq \sup_{\Delta \in \mathcal{R}_\theta} |x^\top\Delta| + \sigma_c.$$

Different verifier shapes correspond to different support functions $\sup_{\Delta \in \mathcal{R}_\theta} |x^\top\Delta|$.

**(1) Ellipsoidal verifier.**    Consider the anisotropic ellipsoid

$$\mathcal{R}_\theta = \big\{\theta : (\theta - \theta_c)^\top A(\theta - \theta_c) \leq r^2\big\}, \qquad A \succ 0.$$

Let $\Delta = \theta - \theta_c$. Changing variables $\Delta = A^{-1/2}u$ with $\|u\|_2 \leq r$ yields

$$\sup_{\Delta^\top A\Delta \leq r^2} |x^\top\Delta| = r\|A^{-1/2}x\|_2 = r\sqrt{x^\top A^{-1}x}.$$

Thus the acceptance condition becomes

$$|y - x^\top\theta_c| \leq r\sqrt{x^\top A^{-1}x} + \sigma_c.$$

**(2) Polyhedral $\ell_1$ verifier.** For the $\ell_1$ knowledge region

$$\mathcal{R}_\theta = \{\|\theta - \theta_c\|_1 \le r\},$$

the perturbation satisfies $\|\Delta\|_1 \le r$. Using Hölder duality,

$$\sup_{\|\Delta\|_1 \le r} |x^\top \Delta| = r\|x\|_\infty.$$

The corresponding acceptance rule is

$$|y - x^\top \theta_c| \le r\|x\|_\infty + \sigma_c.$$

Although ellipsoidal and $\ell_1$ (polyhedral) regions induce different forms of acceptance sets, both yield the same qualitative retraining behavior: $\hat{\theta}^{(k)}$ *consistently move toward the verifier center* $\theta_c$. The empirical trajectories under both shapes are shown in Figure 8.

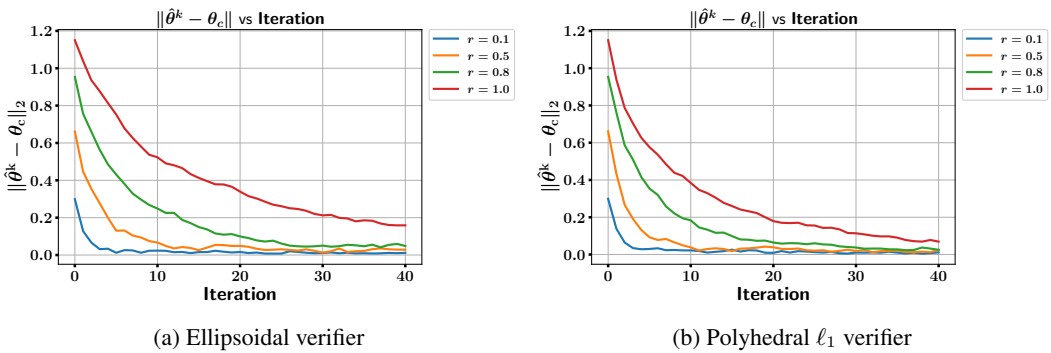

(a) Ellipsoidal verifier                (b) Polyhedral $\ell_1$ verifier

Figure 8: Retraining trajectories under two different verifier shapes. In both cases, $\hat{\theta}^{(k)}$ empirically converges toward the verifier center $\theta_c$.

### E.3  MNIST-SPECIFIC FID EVALUATION

The standard Fréchet Inception Distance (FID) is widely used in generative modeling, including on MNIST, following prior work such as Dai & Wipf (2019); Leontev et al. (2020); Chan & Sithungu (2024). Nonetheless, we agree that Inception embeddings are not tailored to handwritten digits and may not fully capture perceptual similarity on MNIST.

To address this point, we introduce a **MNIST-specific FID** variant. We train a lightweight convolutional network directly on MNIST classification, and compute FID using the penultimate-layer activations as the embedding space. This produces a domain-appropriate FID measure while preserving the same statistical structure as the original metric. These results confirm that our conclusions are robust to the choice of embedding and do not depend on the use of vanilla FID.

**Results.** Figures 9a and 9b report the new FID scores under our retraining framework for all verifier sizes. Consistent with the standard FID curves in the main paper.

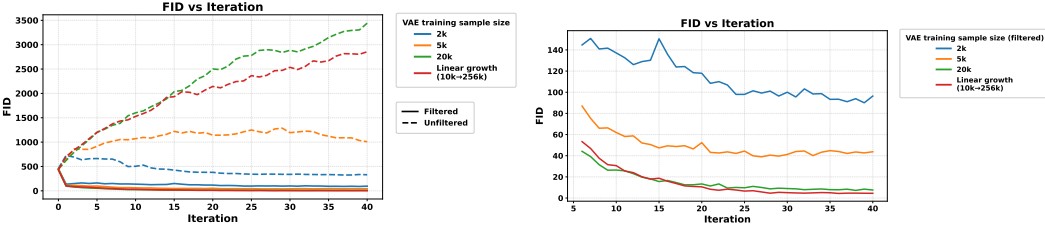

(a) **MNIST-specific FID over retraining iterations.**         (b) **Zoom-in view for filtered runs.**

Figure 9: **MNIST-specific FID using our MNIST-trained feature embedding.** Both results confirm that our conclusions remain unchanged when replacing standard FID with a domain-specific metric.

### E.4    DIFFERENT INITIAL SAMPLE SIZES

We assess the robustness of verifier-guided retraining by varying the number of real MNIST images used to train the initial CVAE ($1k, 2k, 3k, 4k, 60k$). For small and medium initial sample sizes, verifier filtering yields clear early FID improvements and then stabilizes performance, whereas unfiltered retraining quickly degrades. When the generator is initialized on all 60k real images, verifier filtering no longer improves FID over the initial model, but it still effectively prevents the severe collapse observed under unfiltered retraining.

We perform our main experiments in a low–real-data regime (e.g., with 500 initial images), where the verifier, having been trained on a much larger subset of MNIST, holds strictly more external knowledge than the generator. According to our theory, this is exactly the regime in which verifier-guided retraining should provide true improvement rather than simple stabilization, because the verifier contributes additional information. In contrast, when the generator is initialized on the full 60k training images, the verifier would need access to an even stronger source of external information to achieve improvement; otherwise it can only prevent collapse. For this reason, the small initial sample size serves as the most informative regime for highlighting the verifier's knowledge-injection effect and demonstrating the improvement phenomena predicted by our theoretical framework.

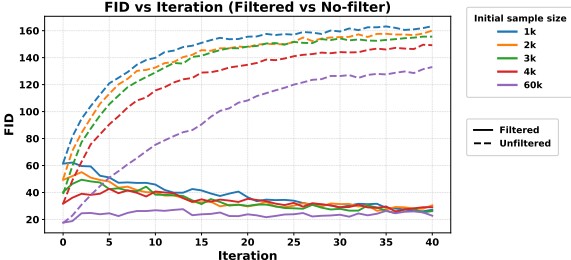

Figure 10: FID across retraining iterations under different initial sample sizes, comparing verifier-filtered and unfiltered synthetic retraining.

# F    USE OF LARGE LANGUAGE MODELS

The authors acknowledge the use of ChatGPT for assistance in improving plot figures, as well as for checking grammar and spelling. All scientific contributions, analyses, and interpretations are solely the work of the authors.

