# OpenReview forum: "Escaping Model Collapse via Synthetic Data Verification:  Near-term Improvements and Long-term Convergence"
_ICLR.cc/2026/Conference — ICLR 2026 Poster_

### Official Review · Reviewer_ENpW · 2025-10-30

**Soundness:** 3
**Presentation:** 3
**Contribution:** 2
**Rating:** 6
**Confidence:** 4

**Summary:**

The paper studies verifier-guided synthetic retraining: a loop that (i) generates synthetic data from the current model, (ii) filters it with an external verifier via a binary accept/reject rule, and (iii) retrains only on the verified subset. In a linear-regression setting, the authors give a one-step MSE bound that separates synthetic variance reduction from verification bias/variance. They then analyze the multi-round process and prove it behaves as a noisy contraction that converges to the verifier’s knowledge center. Experiments in linear regression and a CVAE on MNIST show early, clear gains with verification; long-run plateau depended on verifier quality.

**Strengths:**

- The work formalizes a widely used practice, i.e., filtering synthetic data with a verifier and shows how it changes collapse dynamics.

- A clear theoretical framework and detailed analysis followed by empirical evidence in simple setup.

**Weaknesses:**

- Core analysis is linear regression with a spherical verifier and a special synthetic design; extensions to non-linear models are discussed but not derived. This limits direct transfer of the rates/conditions to modern LMs/vision models.

- MNIST/CVAE is a clean demo but dated; no language or large-scale vision results. Also, while FID trends are solid, downstream task metrics or human evals would strengthen claims about generative quality.

- In the MNIST setup, the useful verifiers are trained with much more training data than the initial 500 data points for the VAE. In the reality, for images, we will normally use all data possible to train both the generative model and the verifier, therefore, another more practical setup is to fix the training set for both the original VAE and the verifier, and iterate from that.

**Questions:**

- How sensitive are the results to the shape of the verifier region?

- A probabilistic perspective might also be interesting since these are generative models. The retraining process can be thought of as to move the original training data distribution towards the distribution defined by the verifier, which actually defines a generative model itself. When the retraining iteration goes to infinite, we are distilling the generative model defined by the verifier into a parametric generative model e.g., a VAE.

---

> ### Author Response · Authors · 2025-11-21
>
> We thank the reviewer for appreciating the value of our work. We respond to the reviewer’s criticisms mentioned in weaknesses and answer the raised questions in the following.
>
> ###  **Q1: Sensitivity to the shape of the verifier region**
>
> Our near-term bias–variance trade-off and long-run convergence is *not sensitive* to the verifier region shape.  In the proof of Theorem 4.1, we effectively show that for a given set of orthogonal design directions $\{v_1, \ldots, v_p\}$, each projected coordinate $v_j^\top \theta^{k}$ evolves as a one-dimensional process whose stationary point is the midpoint of the projection of the verifier’s knowledge region onto $v_j$. This remains true even if the region is asymmetric or irregular; hence $\hat\theta^{k}$ still converges because all projections converge. We adopt a spherical region in the main text for tractability and clarity: in that case the limit is $\theta_c$, and the verifier “bias’’ $\|\theta_c - \theta^\ast\|$ has a clean interpretation.
> However, while convergence is guaranteed more broadly, the uniqueness of the limiting point may depend on the verifier’s region shape, especially when the shape is irregular.  When the knowledge set is centrally symmetric (e.g., ball, ellipsoid, polytope with a well-defined center), the intersection of all projection midpoints is unique, so the limit is design-independent.
>
> **For irregular sets, however, the intersection of projection midpoints can depend on the chosen directions $\{v_j\}$**; different orthonormal bases may yield different fixed points. Therefore, for an irregularly shaped knowledge region, a random design of covariate $X^k$ is more ideal, and we believe that under a random design, the convergence point will correspond to the root of a certain expectation equation, which does not depend on any single finite set of directions (though it still depends on the distribution of the random design). This is an interesting direction for future work.
> We also added, in Appendix D.2 of the updated manuscript, two experiments using verifier regions of different shapes (ellipsoidal and polyhedral). In both cases, the retraining estimator still converges toward the verifier’s knowledge center, demonstrating that our conclusions are robust to the choice of verifier shape.
> ### **Q2. Probabilistic Perspective**
> We thank the reviewer for pointing out the probabilistic viewpoint. Although our analysis is carried out at the level of parameter estimation, the same dynamics admit a natural distributional interpretation. In our linear regression setup, the (conditional) data distribution effectively moves from  $p(y \mid x) = \mathcal{N}(x^\top \theta^\star, \sigma^2)$ toward $p_V(y \mid x) = \mathcal{N}(x^\top \theta_c, \sigma^2)$; that is, the verifier induces a new “target’’ conditional distribution centered at $\theta_c$, and iterative retraining distills this verifier-induced distribution into the parametric model.
> Our proof of Theorem 3.1 and 4.1 exploits two key ingredients: (1) the conditional mean $x^\top \theta$ is linear in $\theta$; and (2) under Gaussian noise, the verifier-induced truncated distribution admits closed-form moments, so the update $\hat{\theta}_{k+1}$ is a contraction toward $\theta_c$, which is analytically tractable and leads directly both to our finite-sample MSE characterization in Theorem 3.1 and to our description of how the convergence rate depends on the synthetic sample size $n$ in Theorem 4.1.
> Extending this analysis to other conditional distributions (e.g., logistic or Poisson regression, more general exponential-family models, or latent-variable generative models) would certainly be interesting, but would require additional analytical tools to obtain finite-sample convergence characterizations comparable to ours. We therefore view this as a promising direction for future work.

---

> > ### Author Response · Authors · 2025-11-21
> >
> > ###  **W1: Special synthetic design**
> > Thank you for raising this point. The assumption that synthetic covariates align with a fixed orthonormal basis is used primarily for analytical tractability. It removes rotational variability and cleanly decouples the dynamics along singular directions, but it is **not essential** to the bias–variance trade-off or long-run convergence behavior.
> >
> > Intuitively, this design can be relaxed. If we write $X = R V$, where $V$ contains the right singular vectors of the real design matrix, then rotating to any other orthonormal basis $V_1$ or $V_2$ yields estimators that differ only by a change of coordinates; without verifier filtering, the retraining dynamics are therefore invariant to the chosen basis. With verifier filtering, the proof becomes more involved because acceptance depends on the alignment between synthetic and real data rather than the coordinate system, but we expect the same invariance to hold.
> >
> > To support this, we added new simulations in Appendix D.1 using **random synthetic designs** (e.g., i.i.d. Gaussian covariates). These experiments show the **same qualitative retraining behavior**, confirming that the orthonormal design is a convenient but non-essential modeling simplification.
> >
> > ###  **W2: MNIST/CVAE additional evaluation**
> >
> > We agree that MNIST is a small-scale domain. Our goal was to empirically validate the mechanisms predicted by our theory in a clean environment. In Appendix D.4 of the revised manuscript, we provide qualitative samples that visibly demonstrate the improvement achieved by filtered retraining. In particular, filtered retraining clearly enhances generation quality, whereas unfiltered retraining with the same synthetic sample size leads to collapse. We further show that the final performance depends on verifier quality (strong verifiers drive further improvement, while weak verifiers can induce degradation) consistent with the quantitative trends observed in our FID results. These examples complement the FID curves and illustrate the exact improvement pattern predicted by our theory. While our current focus is to establish the theoretical mechanisms in the analyzable setting, extending the analysis and experiments to more complex model is a promising direction that we plan to pursue in future work.
> >
> > ### **W3: Use all data for both generator and verifier?**
> >
> > We would like to clarify our experimental choice of training the verifier on more real data than the generator. Our theory shows that verifier-based retraining improves the model only when
> > * The verifier contains **external knowledge** not available to the generator.
> > * This knowledge has sufficiently small bias so that the bias–variance trade-off is favorable.
> >
> > This setup also reflects practical real-world scenarios—common among small and medium-sized businesses or data-scarce domains such as healthcare—where one often has very little real data to train the generator but access to a reasonably strong external verifier, such as medical experts.
> >
> > In our setup, the verifier is a discriminator; its “external knowledge’’ is exactly the additional real images it is trained on. If the generator and verifier are trained on the same real dataset, then the verifier has **no extra information** to contribute.
> > As predicted by our theory, we added new results showing that when using all data to train both the generative model and the verifier **(Figure 10, Appendix D.5)**, the verifier’s contraction effects persist and iterative training converges, but no improvement beyond the original generator occurs.
> >
> > Starting from a weak generator trained on 500 real images allows us to clearly demonstrate the verifier’s knowledge-injection effect and the resulting improvements. If we instead begin with a generator trained on the full MNIST dataset, then—since the verifier cannot access any additional real data beyond what was already used for training—our framework predicts that no further improvement is theoretically possible. We have conducted this experiment and included the results in Appendix D.5 of the revised manuscript, which empirically confirm this prediction.

---

> ### Author Response · Authors · 2025-12-02
>
> ### **W2: lack of language or large-scale vision results**
>
> To address reviewer's concern, we conducted additional experiments on a large-scale news-summarization task using the `XSUM` dataset and the modern LLM `SmolLM2-135M`. We compared retraining with **ROUGE-1–filtered synthetic data** versus **unfiltered synthetic data**. The results show that  (1) filtering synthetic data consistently improves model performance, and (2) performance converges as the number of training iterations increases. In contrast, retraining on unfiltered synthetic data yields no measurable improvement. These findings demonstrate that our theoretical conclusions **generalize to large-scale datasets and contemporary LLMs**. Further details are provided in **Appendix D.6** of the updated PDF.

---

### Official Review · Reviewer_TG7U · 2025-10-30

**Soundness:** 3
**Presentation:** 3
**Contribution:** 2
**Rating:** 2
**Confidence:** 4

**Summary:**

This paper investigates the effect of verifier quality on model collapse from training on self-generated synthetic data.  The authors structure their analysis from linear regression with a verifier that has controllable bias.  It is discovered that a mildly biased verifier may provide initial improvements, but will ultimately lead to performance plateaus or collapse over many iterations.  The authors evaluate their work on linear regression as well as preliminarily on a CVAE implementation on MNIST.

**Strengths:**

The paper is quite clearly presented, and the math is approachable to verify, being based on linear regression.

**Weaknesses:**

The paper is quite toy from the perspective of model (linear regression), data, and verifier.  This may be fine for theoretical proofs, but it does limit the scope of takeaway when considering the applicability to frontier generative models.  It would be interesting to see actual experiments on language modeling and large-scale pretrained models beyond such toy settings.  Particularly as there are a variety of reinforcement learning from verifiable rewards works on frontier models at the moment, it should not be difficult to apply it for the verifier-filtered experiments proposed here.

Even MNIST is a bit toy compared to available datasets of natural images.

The takeaways also make intuitive sense, rendering the findings a bit unsurprising - one does expect that a mildly biased verifier should help initially but performance should not be able to break free from the biases of the verifier when the scale of seen data is predominantly that filtered by the verifier.

**Questions:**

Why is verifier-based filtering in particular important to study thoroughly - even for frontier generative models?  Whereas naively retraining the generative model on self-generated synthetic data may indeed lead to model collapse, the field has seen success in avoiding this through verifiable rewards (Reinforcement Learning from Verifiable Rewards, as used in DeepSeek [1], Tulu3 [2]).  Why do filtered retraining from a verifier rather than use all data but labeled with a reward from the verifier (such as labelling unsuccessful generations with a 0)?  The assumption for the components is the same in both approaches, but RLVR seems to work better than verifier-based filtered training.

[1] DeepSeek-AI et al., DeepSeek-R1: Incentivizing Reasoning Capability in LLMs via Reinforcement Learning, 2025.

[2] Lambert et al., Tulu 3: Pushing Frontiers in Open Language Model Post-Training, 2024.

---

> ### Author Response · Authors · 2025-11-22
>
> We thank the reviewer for appreciating the theory and presentation of our work and for the thoughtful comments, particularly for pointing out the comparison with the reinforcement learning with verified rewards (RLVR) paradigm. We believe the reviewer’s criticism of our paper might stem from misperception about the applicability of RLVR. We hope our clarification below could help clarify the misperception, and are eager to engage with the reviewer further if any additional elaboration is needed during the rebuttal.
>
> ###  **On “Toy Setting”**
>
> As the reviewer also appreciated, our paper’s focus is indeed on principled understanding of an already adopted practical training pipeline, i.e., verifier-based filtering of synthetic data for model retraining. We respectfully point out that making theoretical advancement in this direction is not an easy feat. Indeed, the “toy” linear regression framework we study has been a standard formal model in a line of recent published papers on model collapse[1-4]. It appears a bit unfair to undermine our setting’s merit, given that it follows a rich and active line of recent papers in the past year.
>
> ### **On “The takeaways are intuitive, rendering the findings a bit unsurprising”**
>
> This general comment has multiple facets – while some facets may be true, others are not. By “takeaways”, if the reviewer means the very high-level message that “verified synthetic data could possibly improve re-training”, then this indeed is not “surprising” – in fact, some practical success already demonstrated this. However, the value of principled study is to dive into its underlying rationale to answer questions such as (1) does this high-level message always hold? (2) what factors drive its success or failure? And (3) how to quantify the tensions among these factors? We believe answers to these questions are **not intuitive** – this is precisely what our theoretical analysis tries to analyze and, notably, obtaining answers to these questions are not easy. In a similar vein, while it is “intuitive” or “unsurprising” that more data will lead to more accurate model predictions, the celebrated theoretical framework of *Probably approximately correct (PAC) learning* is still valuable (awarded Turing Award in 2010), because it helps answer questions of the above theoretical nature in the PAC-learning framework.
>
> To re-iterate our theoretical findings, our **Theorem 3.1 gives a finite-sample MSE bound** that quantifies the tension between the verifier’s accuracy, the strictness of the verification rule, and the synthetic sample size; moreover, it matches well with empirical simulations (updated Figure 1). **Theorem 4.1 further quantifies how the convergence rate depends on synthetic data sample size**, revealing contraction dynamics. We respectfully point out that these quantitative results should not be viewed as “intuitive” or “unsurprising”.

---

> > ### Author Response · Authors · 2025-11-22
> >
> > ###  **Q1. Why is it well-motivated to study verifier-based filtering, while not  RLVR?**
> > While indeed being a useful approach,  **RLVR is only applicable in settings where reward signals are clearly verifiable** – these are mostly reasoning tasks with deterministic outcomes such as math and coding [6,7,8]. However, **our study is motivated by, and helps shed insights on, many important training stages and domains that do not have clearly verifiable rewards**, such as:
> >
> > - **Pre-training:** RLVR is mostly used for fine-tuning. There is no established or scalable way to apply RLVR in pre-training, where reward signals are ambiguous or unavailable.
> >
> > - **Model alignment:** Even alignment procedures do not typically use RLVR but rely on pairwise comparison (e.g., RLHF[5]), precisely because assigning absolute reward values is difficult and noisy.
> >
> > - **Generative domains without clear metrics**: For example, evaluating image quality or open-ended language generation with a scalar reward is highly subjective. **In contrast, binary accept/reject filtering is much less noisy and widely used in practice.**
> >
> > In addition, **verifier-based filtering represents a core primitive that already achieves success in frontier LLM training**(e.g., the pre-training of Deepseek-Coder [8].), including those that may also incorporate RLVR at later stages. Well-recognized practical advantages of this “generate–verify–retrain” is that it plugs directly into all training pipelines and is stable and scalable. Given its success and advantages, we do believe it is valuable to conduct principled analysis for this fundamental mechanism. This is also justified by the rich line of recent research on this topic in the past two years (see Related Work section of our draft).
> >
> > ###  **Q2. Why filtered retraining instead of labeling all data with rewards?**
> > This is because in many settings (e.g., domains without clearly verified rewards or supervised fine-tuning using text corpus), it is either too costly to obtain reward signals or the reward signals are too noisy. The reviewer’s question could be asked similarly for the seminal  *Reinforcement Learning from Human Feedback* (RLHF) approach [5] that enabled the success of ChatGPT – why do OpenAI train the InstructGPT model using comparative feedback by human labelers, while not directly asking human labelers to score the feedback? The answer is the same.

---

> ### Author Response · Authors · 2025-11-22
>
> ### **Additional Experiments on News Summarization Tasks**
> We sincerely appreciate the reviewer’s suggestion to evaluate our method on larger-scale experiments. In response, we have begun running additional experiments on the XSUM news-summarization dataset [9], a widely used benchmark in natural language processing. Our experimental procedure is as follows:
> - Step 0: Load the XSum dataset, which contains 204,045 training samples and 11332 test samples.
> - Step 1: Fine-tune HuggingFaceTB/SmolLM2-135M-Instruct [10] on 12.5% of the training set for one epoch, and evaluate its ROUGE-1 score on the test set. Implementation details are listed below.
> - Step 2: Use the fine-tuned model to generate synthetic summaries for all 204,045 training examples.
> - Step 3: Apply an **oracle filter** to select the top 12.5% synthetic article summary pairs (x, y') based on the ROUGE-1 score between the ground-truth summary y and the generated synthetic summary y’.
> - Step 4: Retrain the model on these selected synthetic examples and evaluate its ROUGE-1 score on the test set.
> - Step 5: Repeat Steps 2–4 for multiple iterations. Keep track of ROUGE-1 score on the test set until the performance stabilizes.
>
> **Implementation details:**
>
> **Prompt**: f“Article: {article}. Summary: {summary}”
>
> **Fine-tuning and generating details.** We keep our experiment design similar to [11]. Throughout all phases of evaluation and generation, we employ greedy decoding. Given that news summarization is a low-entropy task, greedy decoding is chosen to ensure quality generation. Consistent with common practice, fine-tuning is limited to a single epoch. Throughout the experiments, all the fine tuning is with full parameter tuning to better capture the scaling law
>
> For training the generator, SmolLM2-135M-Instruct, a current state-of-art small model, we set the learning rate to 5e-5, the learning rate scheduler as ‘cosine’, the number of epochs to 1, the total batch size to 32, and the others to the default value. However, to shorten the training time, we set the block size to **256**.
>
> For generating the synthesized data, we use the greedy strategy to generate a summarization for each news in the training set. For evaluation, we first use greedy strategy to generate a summarization for each news in the test set, and then calculate the Rouge-1 score between the generated summarization and the corresponding ground truth, and finally report the average of the Rouge-1 scores of all test data. When calculating the perplexity, we only calculate the perplexity for the generated summary.
>
> Computational Resources. All experiments were conducted using a dedicated computational cluster equipped with 4 NVIDIA A800 GPUs, each with 256 GB of memory. Our training and inference processes are performed on the cluster.
>
> Due to time constraints, we currently only have partial results. In the first five rounds of retraining using the oracle filter, we observe consistent improvement at each iteration, which aligns with our theoretical prediction that a high-quality verifier enhances model performance through synthetic retraining. We will continue running additional retraining iterations and evaluate the effects of varying verifier quality. **The remaining results will be updated once the ongoing experiments are completed.**
>
> | Iteration | 0 | 1 | 2 | 3 | 4 |
> |-----------|---|---|---|---|---|
> | ROUGE-1   | 0.1451 | 0.1623 | 0.1639 | 0.1648 | 0.1652 |

---

> > ### Author Response · Authors · 2025-11-22
> >
> > References:
> >
> > [1]Elvis Dohmatob, Yunzhen Feng, and Julia Kempe. Model collapse demystified: The case of regression. Advances in Neural Information Processing Systems, 37:46979–47013, 2024a.
> >
> > [2]Elvis Dohmatob, Yunzhen Feng, Arjun Subramonian, and Julia Kempe. Strong model collapse. In The Thirteenth International Conference on Learning Representations, 2025.
> >
> > [3]Matthias Gerstgrasser, Rylan Schaeffer, Apratim Dey, Rafael Rafailov, Tomasz Korbak, Henry Sleight, Rajashree Agrawal, John Hughes, Dhruv Bhandarkar Pai, Andrey Gromov, et al. Is model collapse inevitable? Breaking the curse of recursion by accumulating real and synthetic data. In First Conference on Language Modeling (COLM), 2024.
> >
> > [4]Xuekai Zhu, Daixuan Cheng, Hengli Li, Kaiyan Zhang, Ermo Hua, Xingtai Lv, Ning Ding, Zhouhan Lin, Zilong Zheng, and Bowen Zhou. How to synthesize text data without model collapse? Forty-second International Conference on Machine Learning, ICML, 2025.
> >
> > [5]Long Ouyang, Jeffrey Wu, Xu Jiang, Diogo Almeida, Carroll Wainwright, Pamela Mishkin, Chong Zhang, Sandhini Agarwal, Katarina Slama, Alex Ray, et al. Training language models to follow instructions with human feedback. Advances in neural information processing systems, 35:27730–27744, 2022.
> >
> > [6]Qiying Yu, Zheng Zhang, Ruofei Zhu, Yufeng Yuan, Xiaochen Zuo, Yu Yue, Weinan Dai, Tiantian Fan, Gaohong Liu, Lingjun Liu, et al. Dapo: An open-source llm reinforcement learning system at scale. arXiv preprint arXiv:2503.14476, 2025
> >
> > [7]Daya Guo, Dejian Yang, Haowei Zhang, Junxiao Song, Peiyi Wang, Qihao Zhu, Runxin Xu, Ruoyu Zhang, Shirong Ma, Xiao Bi, et al. Deepseek-r1 incentivizes reasoning in llms through reinforce- ment learning. Nature,
> >
> > [8]D Guo, Q Zhu, D Yang, Z Xie, K Dong, W Zhang, G Chen, X Bi, Y Wu, YK Li, F Luo, Y Xiong… DeepSeek-Coder: When the Large Language Model Meets Programming--The Rise of Code Intelligence arXiv preprint arXiv:2401.14196, 2024
> >
> > [9] Shashi Narayan, Shay B. Cohen, and Mirella Lapata. Don’t give me the details, just the summary! Topic-aware convolutional neural networks for extreme summarization. In Proceedings of the 2018 Conference on Empirical Methods in Natural Language Processing, Brussels, Belgium, 2018.
> >
> > [10] Loubna Ben Allal, Anton Lozhkov, Elie Bakouch, Gabriel Martín Blázquez, Guilherme Penedo,Lewis Tunstall, Andrés Marafioti, Hynek Kydlíˇcek, Agustín Piqueres Lajarín, Vaibhav Srivastav,Joshua Lochner, Caleb Fahlgren, Xuan-Son Nguyen, Clémentine Fourrier, Ben Burtenshaw, Hugo Larcher, Haojun Zhao, Cyril Zakka, Mathieu Morlon, Colin Raffel, Leandro von Werra, and Thomas Wolf. Smollm2: When smol goes big – data-centric training of a small language model, 2025. URL https://arxiv.org/abs/2502.02737.
> >
> > [11] Yunzhen Feng, Elvis Dohmatob, Pu Yang, Francois Charton, and Julia Kempe. Beyond model collapse: Scaling up with synthesized data requires reinforcement. In ICML 2024 Workshop on Theoretical Foundations of Foundation Models, 2024.

---

> ### Author Response · Authors · 2025-12-02
>
> ### **Update on Large-Scale LLM Experiments**
>
> We updated the News Summarization task results by running 15 rounds of synthetic retraining and reporting the ROUGE-1 score across all rounds for both the filtered and unfiltered conditions. In the unfiltered condition, we retrain on the same number of synthetic examples as in the filtered condition to ensure a fair comparison. The filtered retraining procedure produces a consistent, **monotonic improvement over the first several iterations before stabilizing**, aligning with our theoretical predictions. In contrast, the unfiltered baseline shows no meaningful improvement and instead fluctuates around its initial performance, demonstrating that synthetic retraining without quality control fails to provide performance gains. These results further show that our theory holds in large-scale settings, where the dynamics of synthetic retraining closely match our theoretical predictions. Please refer to Appendix D.6 for additional details.

---

### Official Review · Reviewer_j2Ts · 2025-10-31

**Soundness:** 3
**Presentation:** 3
**Contribution:** 3
**Rating:** 6
**Confidence:** 4

**Summary:**

This paper mainly studies the role of verification in preventing model collapse using a linear regression framework. Their analysis finds that, in the short term, the proper use of verification can enhance performance by reducing variance. However, in the long term, it may cause the model parameters to converge toward those used to construct the verifier. The results of this paper are supported by rigorous theoretical analysis and experimental evidence.

**Strengths:**

The paper is clearly written and easy to follow.

The claims in the paper are supported by rigorous theoretical analysis and comprehensive experiments.

The results are quite intuitive and make a lot of sense.

**Weaknesses:**

I think this paper presents a solid theoretical work. Below, I will list some drawbacks; however, they do not constitute reasons for rejection, as we all understand that in theoretical analysis, it is often challenging to address more complex but practical models.

1. During iterations, it is assumed that the synthetic covariates can only be the copies of a fixed orthonormal set. This assumption might be a little bit strong, can this be further relaxed?

2. It seems that the overparametrized regime has not been considered. Is it possible to generalize the results to the overparametrized regime?

3. In practice, the verifier may evolve over time. Could the authors consider whether a similar analysis can be conducted under this setting?

**Questions:**

Please see the weaknesses above.

---

> ### Author Response · Authors · 2025-11-21
>
> We are grateful to the reviewer for the constructive and insightful suggestions. The points raised indeed open up several valuable avenues for generalizing our theoretical framework.
>
> ###  **W1. Synthetic covariates restricted to a fixed orthonormal set**
>
> Yes, the assumption can indeed be relaxed. We clarify below.
>
> **Why we used the orthonormal design in the main theory:** the assumption that synthetic covariates align with a fixed orthonormal basis is used primarily for analytical tractability. It removes rotational variability and cleanly decouples the dynamics along singular directions, but it is **not essential** to the bias–variance trade-off or long-run convergence behavior.
>
> **Intuition:  this assumption can be relaxed because the choice of basis does not affect retraining dynamics:** If we write $X = R V$, where $V$ contains the right singular vectors of the real design matrix, then rotating to any other orthonormal basis $V_1$ or $V_2$ yields estimators that differ only by a change of coordinates; without verifier filtering, the retraining dynamics are therefore invariant to the chosen basis. With verifier filtering, the proof becomes more involved because acceptance depends on the alignment between synthetic and real data rather than the coordinate system, but we expect the same invariance to hold.
>
> To support this, we added new simulations in Appendix D.1 using **random synthetic designs** (e.g., i.i.d. Gaussian covariates). These experiments show the **same qualitative retraining behavior**, confirming that the orthonormal design is a convenient but non-essential modeling simplification.
>
> ###  **W2. Extension to the overparameterized regime**
>
> We agree that extending the results to the overparameterized setting $p>n$ is indeed possible, for instance, using lasso estimators under the same verifier mechanism. While this extension would require additional technical work, we see this as a compelling direction for future work.
>
> ###  **W3. Evolving verifier over iterations**
>
> We appreciate this interesting suggestion. This extension can be explored in the linear regression setting by allowing the verifier’s center $\theta_c$ to evolve over time. Our contraction-based picture naturally extends, at an intuitive level, to this setting. Here is our current idea:
> Let $\hat \theta_{c,k}$ be the verifier knowledge center at round k(it may evolve based on some other external knowledge).  Roughly we may write $\hat\theta_{k+1} \approx (1-\rho)\\theta_{c,k} + \rho\\hat\theta_k$,  $0 < \rho < 1$, where $\rho$ is the contraction factor identified similarly in the fixed-verifier analysis. Let model error $e_k := \hat\theta_k - \theta^\star$ and verifier bias $ b_k := \theta_{c,k} - \theta^\star$.
>
> Then the update becomes $$e_{k+1} \approx (1-\rho)\theta_{c,k} + \rho\hat\theta_k - \theta^\star = \rho e_k + (1-\rho)b_k.$$ This is the key linear recursion. Solving it (ignoring noise for intuition) gives $$e_k= \rho^k e_0  + (1-\rho)\sum_{t=0}^{k-1} \rho^{\,k-1-t} b_t.$$ Thus $\hat\theta_k$ behaves as an **exponential moving average of the verifier centers** $\theta_{c,k}$. This yields the following intuitive conclusions:
>
> * If the evolving verifier is itself consistent, $b_k \to 0$ (i.e., $\theta_{c,k} \to \theta^\star$), then $e_k \to 0$ and the model converges to the true parameter.
>
>  * If $\theta_{c,k}$ converges to a biased limit $\theta_\infty \neq \theta^\star$, then $\hat\theta_k$ converges to the same limit $\theta_\infty$: the model tracks the verifier’s asymptotic bias.
>
> While this recursion is approximate and omits finite-sample noise, it illustrates that our contraction-based analysis extends naturally to time-varying verifiers and we think the estimator will still converge to the limit of the verifier center(if the verifier converges). A more formal analysis of this evolving verifier is an exciting direction for future work.

---

> > ### Comment · Reviewer_j2Ts · 2025-11-23
> >
> > Thank you for the rebuttal.
> >
> > I will keep my positive score.

---

### Official Review · Reviewer_SkJA · 2025-11-01

**Soundness:** 3
**Presentation:** 4
**Contribution:** 2
**Rating:** 6
**Confidence:** 4

**Summary:**

This paper studies the role of verifier-based filtering in retraining on generated data to avoid model collapse. It shows short-term variance reduction and long-term convergence toward the verifier’s knowledge center.

**Strengths:**

- The paper is clearly written and easy to follow.
- Verification plays an important role in preventing collapse in self-consuming models. It’s a simple and intuitive idea.

**Weaknesses:**

- I am mainly concerned about where the incremental theoretical contribution lies. The connection to prior verification in self-consuming works is not discussed in depth (see Question 1).
- The paper considers pure synthetic replacement. In practice, retraining typically mixes in real data or accumulates data.
- Experiments are small-scale (MNIST/CVAE only) and lack validation on larger datasets or modern models.

**Questions:**

1. The paper briefly cites [1] but does not clearly explain how it differs from that line of work. Could the authors clarify what the theoretical novelty is relative to [1][2], and how this work connects to recent studies [3][4]?
2. I have some concerns about experimental setup:
    - Does initializing on only 500 real MNIST images limit the baseline and exaggerate the apparent early gains?
    - The paper states that "the verifier is trained on varying amounts of real data together with an equal number of synthetic samples." Which round’s synthetic samples are used? Is the verifier retrained each iteration (i.e., changing over time), or fixed?
    - The paper mentions keeping the Top-$10\%$ of generated samples. If the verifier is binary, why is there a "top" ranking rather than simply using all passing samples (and reporting number)? If it is deterministic top-scoring selection, does this still match the theory's assumption?

[1] Damien Ferbach, Quentin Bertrand, Joey Bose, and Gauthier Gidel. Self-consuming generative models with curated data provably optimize human preferences. In The Thirty-eighth Annual Conference on Neural Information Processing Systems, 2024.

[2] Xiukun Wei and Xueru Zhang. Self-consuming generative models with adversarially curated data. In Forty-second International Conference on Machine Learning, 2025.

[3] Kareem Amin, Sara Babakniya, Alex Bie, Weiwei Kong, Umar Syed, and Sergei Vassilvitskii. Escaping collapse: The strength of weak data for large language model training. In Scaling Self-Improving Foundation Models without Human Supervision, 2025.

[4] Xuekai Zhu, Daixuan Cheng, Hengli Li, Kaiyan Zhang, Ermo Hua, Xingtai Lv, Ning Ding, Zhouhan Lin, Zilong Zheng, and Bowen Zhou. How to synthesize text data without model collapse? In Forty-second International Conference on Machine Learning, 2025.

---

> ### Author Response · Authors · 2025-11-21
>
> We thank the reviewer for appreciating the value of our work. We respond to the reviewer’s criticisms mentioned in weaknesses and answer the raised questions in the following.
>
> ###  **W1 & Q1. Relation to prior self-consuming / verification works**
>
> [1,2], and our work all study how filtering affects synthetic retraining, which refers to the iterative process where a model generates synthetic data and updates itself using filtered samples, but we address different theoretical questions. [1] and [2] analyze filtering in a **preference-maximization** framework: synthetic samples are curated by a reward model, and their population-level analysis shows that, with infinite curated data, the learned model collapses toward highest-reward regions. In contrast, we work in a linear model $p(y \mid x) = \mathcal{N}(x^\top \theta, \sigma^2)$ and take a **parameter-estimation** viewpoint: our goal is to recover the ground-truth parameter $\theta^\star$ and to quantify how verifier-guided filtering changes the estimator’s bias and variance.
>
> Technically, [1] and [2] optimize expected objectives in the **population level** and do not track finite-sample error, whereas our Theorem 3.1 explicitly characterizes how the estimator’s MSE changes with the sizes of both the real and synthetic datasets, and Theorem 4.1 shows how the convergence rate and limit point depend on the synthetic sample size and verifier accuracy. Such **finite-sample** MSE guarantees and a precise description of how the verifier’s accuracy, the strictness of the verification rule, and the synthetic sample size shapes the long-run behavior of synthetic retraining, which is not captured in [1,2].
>
> [3] assumes access to a **strong “quality function”** that can accurately assess whether synthetic samples are good, and a weak labeler (the generator) that produces these samples. Their focus is therefore on how the external labeler contributes to learning under an already reliable filtering mechanism. In contrast, our setting is from a different perspective: the generator is simply the previous model (i.e., potentially error-prone), and the key driver of improvement or collapse is the quality of the verifier (their “quality function”). Rather than assuming it is good, we analyze how verifier quality impacts retraining dynamics, determining whether synthetic retraining improves, plateaus, or deteriorates performance.
>
> [4] also studies how synthetic data can harm training and proposes a data-editing scheme to mitigate collapse. However, its theory only replaces parts of synthetic data with previous samples to prevent collapse, without showing performance gains, whereas we leverage external knowledge to generate higher-quality synthetic data that **improves the model rather than merely avoiding collapse**.
>
>
> ###  **W2. ​ Pure synthetic replacement vs. mixing real data**
>
> While the consideration of mixing in fresh real data in a verifier-based synthetic retraining is exciting future research direction, we would like to point out that the dynamics and limit behaviour of an accumulating  flow compared to our current replacement flow is essentially the same.
>
> In an accumulating flow at iteration $k+1$, new synthetic data are generated by $\hat\theta_k$, verified, and then combined with previous retrained data. The increment from $\hat\theta_k$ to $\hat\theta_{k+1}$ still depends only on the newly generated (and verified) batch at round $k+1 $.  Hence $\hat\theta_k$ is still a Markov process and admits a transition as Equation (11) in the paper.
>
> More concretely, in the proof of Theorem 4.1, we showed that for fixed orthonormal directions $v_j$, each projected coordinate  $v_j^{\top}\theta^{k}$  evolves as a 1-D contractive update (Equation (27) in the paper). Under accumulating flow, a similar equation can be derived, and the contraction has the form $T(x) = x+\frac{n_k}{(n_1+n_2+... +n_k)} \mathbb{E}[Z \mid a-x<Z<b-x]$, only the rate differs through the weight $n_k/\sum{n_i}$. So the fix point is identical to a replacement flow. One subtlety in an accumulating flow is that the convergence rate and noise accumulating depend on the expansion rate of $n_k$, which might require slightly different techniques to establish the proof.

---

> > ### Author Response · Authors · 2025-11-21
> >
> > ###  **Q2. Does small initialization exaggerate early gains?**
> >
> > We used a small initial real-data sample because this is the regime where our theory predicts that verifier-guided retraining can provide very clear improvement: the generator is undertrained relative to the verifier, which therefore contributes external information the generator does not yet possess. In contrast, when the generator is initialized on the full 60k training images, the verifier would need access to an even stronger source of external information to achieve improvement; otherwise it can only prevent collapse. To address the reviewer’s concern that small baselines may exaggerate early gains, we added new experiments **(Appendix D.5)** varying the initial real-data size from 1k to 60k. The results show that early FID improvements appear in the small-data regime, consistent with our theory, while large-data initialization yields stabilization but no improvement. These findings confirm that the early gains are not artifacts of a weak baseline. We invite the reviewer to read Appendix D.5 for the full results.
> >
> > ### **Q3. Verifier training protocol**
> >
> > In every round $ t$, we train a new verifier on (i) a fixed subset of real MNIST images and (ii) an equal number of synthetic samples generated by the current generator (i.e., the model at round $t$). We then use the round-$ t$ verifier to score the round-$ t$ candidate synthetic set and retain the verified subset for retraining the generator.
> >
> > Although the verifier is retrained each iteration, its **real-data information remains fixed**, since all verifiers are trained on the same fixed pool of real images. Thus, the verifier’s parameters change over time, but the underlying knowledge it injects does not expand.
> >
> > ###  **Q4. Why “top-k%” if the verifier is binary?**
> >
> > Indeed, the theoretical verifier is modeled in a simplified form as a parameter-level knowledge center $(\theta_c, r)$, which induces a stable analytical threshold to allow us to analyze the effect of verifier knowledge in a tractable way. We further note that in our theoretical analysis, the synthetic sample size $n_k$​ corresponds to the post-filtering number of retained samples rather than the number generated before filtering.
> >
> > In practice, however, the verifier’s knowledge comes from real data via a trained discriminator, whose implicit decision boundary is difficult to extract explicitly. The verifier score distribution varies a lot across rounds and digit classes. This makes it challenging to specify a single fixed threshold that remains stable during training. For this reason, we adopt a data-dependent threshold via top-$k\%$ filtering, which adaptively selects samples based on the current verifier score distribution. This approach serves as a practical surrogate for the theoretical acceptance rule and stabilizes empirical retraining.

---

> > > ### Comment · Reviewer_SkJA · 2025-11-26
> > >
> > > Thank you for the rebuttal. However, I still don’t really understand why the theory assumes a fixed verifier (line 133), while in the experiments the verifier evolves together with the generator. I’m not sure if I missed a discussion explaining why a verifier trained on real data and continually changing synthetic data is still equivalent to a fixed "knowledge ball" in the theory.
> > >
> > >  Anyway, I will keep my positive score.

---

> > > > ### Author Response · Authors · 2025-12-01
> > > >
> > > > Thank you for your positive feedback and for pointing out this clarification. We will further elaborate on the evolved verifier.
> > > >
> > > > In the theory, the verifier’s external knowledge modeled as a ball centered at $\theta_c$ is assumed to be fixed. This assumption is preserved in our experiments. Although the verifier is retrained at each iteration, it  is always trained  on  the *same fixed real dataset* (together with newly generated synthetic samples). As a result, its knowledge about the data distribution remains unchanged across iterations.
> > > >
> > > > The seeming discrepancy arises from the practical implementation of the verifier as a discriminator. A discriminator trained against one generator may not be well-calibrated for samples produced by a later generator, so retraining is necessary to keep the verifier correctly expressing its fixed knowledge on newly generated samples. Importantly, this retraining does not introduce new information or evolve the verifier’s beliefs; rather, it serves to maintain alignment between the discriminator and the current generator.
> > > >
> > > > Therefore, while the *parameters* of the discriminator evolves, the *knowledge it injects remains unchanged, consistent with the theoretical assumption.*

---

> ### Author Response · Authors · 2025-12-02
>
> ### **W3: small-scale experiment and lack validation on larger datasets or modern models**
>
> To address reviewer's concern we conducted additional experiments on a large-scale news-summarization task using the `XSUM` dataset and the modern LLM `SmolLM2-135M`. We compared retraining with **ROUGE-1–filtered synthetic data** versus **unfiltered synthetic data**. The results show that (1) filtering synthetic data consistently improves model performance, and (2) performance converges as the number of training iterations increases. In contrast, retraining on unfiltered synthetic data yields no measurable improvement. These findings demonstrate that our theoretical conclusions **generalize to large-scale datasets and contemporary LLMs**. Further detail can be found in **Appendix D.6** in the updated PDF. Further details are provided in **Appendix D.6** of the updated PDF.

---

### Official Review · Reviewer_Lw2s · 2025-11-02

**Soundness:** 3
**Presentation:** 3
**Contribution:** 2
**Rating:** 6
**Confidence:** 4

**Summary:**

Authors analyze the learning dynamic of generative models iteratively retrained on their own data. The novelty comes from the fact that each step, (potentially bad) data are discarded from the retraining loop, using an external verifier.
Authors validate their theory on synthetic experiments and MNIST.

**Strengths:**

The topic is very timely: quantifying how much can be leverage from synthetic data is a central question for generative modelling.

**Weaknesses:**

- Disconnection between the analysis and motivation. As [1, 2, 3], authors analyze a linear regression setting setting, that does significantly differ from the generative model setting (as started by the authors in the conclusion). Did author try to make the analysis in the true generative model setting? Like [4]? Can authors provide justification of why to study this setting? and it is relevant to generative modelling?
- In particular, the proposed setting looks like well-studied semi-supervised learning results [4, 5]?
- Experiments: for Figure 3, authors mentioned they used the FID metric. I di not find any additional comment, so I assume authors use standard vanilla FID. If so, I think this is a mistake for multiple reasons:
    - FID rely on the Inception embedding, that is a standard for natural images: to the best of my knowledge, MNIST images are not considered natural images.
    - In addition, the point of using a embedding is to lower the dimensionality of the data, in order to have a better approximation of the empirical Wasserstein distance
To my knowledge, the best for MNSIT, is to train a classifier from scratch, and take the last layer for the embedding.
- Figure 3b, what is the "verifier training size" in the caption? Is this a way to vary the quality of the verifier?

[1] Elvis Dohmatob, Yunzhen Feng, and Julia Kempe. Model collapse demystified: The case of regression

[2] Elvis Dohmatob, Yunzhen Feng, Arjun Subramonian, and Julia Kempe. Strong model collapse

[3] Elvis Dohmatob, Yunzhen Feng, Pu Yang, Francois Charton, and Julia Kempe. A tale of tails: Model collapse as a change of scaling laws

[4] Damien Ferbach, Quentin Bertrand, Avishek Joey Bose, and Gauthier Gidel. Self-consuming generative models with curated data provably optimize human preferences

[4] Ben-David, Shai, Tyler Lu, and Dávid Pál. "Does Unlabeled Data Provably Help? Worst-case Analysis of the Sample Complexity of Semi-Supervised Learning." COLT. 2008.

[5] Zhou, Z.-H. (2018). A brief introduction to weakly supervised learn­
ing. National Science Review

**Questions:**

- "Suppose each eigenvalue of the design matrix X 0 is ω( n0 )", what is w? do you assume that the eigenvalues are strictly a function w of n0?
- I do not understand the takeaway of Theorem 3.1, especially I do not understand the discussion around Equation 9.
- "This contribution also clarifies a common misconception: even with a perfect verifier (θc = θ ⋆ ) and infinitely many synthetic samples in one iteration, convergence cannot occur in a single step. As
shown in Theorem 3.1, while infinite samples remove the synthetic variance term, the verification bias+variance term persists." I do not understand this comment: if the verifier is perfect, why would the verification bias persist?
- Could authors comment on the conclusion line 299, with the "3 phases" depending on the verifier, isn't it obvious that if the verifier is unbiased, it will help, and it is not a well-suited verifier, then the filtering procedure will not be helpful?

---

> ### Author Response · Authors · 2025-11-21
>
> We thank the reviewer for appreciating the value of our work. We respond to the reviewer’s criticisms mentioned in weaknesses and answer the raised questions in the following.
>
> ### **W1:On the connection between our linear analysis and generative modelling.**
>
> While our theory is phrased at the level of parameter estimation, it admits a natural probabilistic interpretation that is consistent with the generative viewpoint. In our setup, training pairs $(x,y)$ are drawn from a conditional model $p(y \mid x) = \mathcal{N}(x^\top \theta^\star, \sigma^2)$, and the verifier has a different conditional distribution $p_{V}(y \mid x) = \mathcal{N}(x^\top \theta_c, \sigma^2)$ and induce this distribution by filtering out samples that disagree with its “knowledge center’’ $\theta_c$. Iterative generate–filter–retrain therefore moves the effective training distribution from $p(y\mid x)$ toward $p_V(y\mid x)$, and our Theorem 4.1 shows that, at the parameter level, this corresponds to a contraction of $\hat\theta_k$ toward $\theta_c$. In other words, although we analyze $\hat\theta_k$, one can equivalently view the procedure as distilling a verifier-induced conditional distribution into a parametric model.
>
> **We chose this linear setting for two reasons.**  First, it is standard in the literature (including [1,2,3]) to use linear regression as a tractable surrogate for understanding model collapse. Second, our proofs of Theorems 3.1 and 4.1 exploit two key ingredients: (i) the conditional mean $x^\top \theta$ is linear in $\theta$; and (ii) under Gaussian noise, the verifier-induced truncated distribution admits closed-form moments, so the update $\hat{\theta}_{k+1}$ is a contraction toward $\theta_c$, which is analytically tractable and leads directly both to our finite-sample MSE characterization in Theorem 3.1 and to our description of how the convergence rate depends on the synthetic sample size $n_k$ in Theorem 4.1. In contrast, [4] conducts a population-level analysis that requires infinitely many synthetic samples, whereas our results explicitly show how verifier-guided retraining’s performance depends on the synthetic sample size at each iteration.
>
>
> ### **W2. Relation to Semi-Supervised Learning.**
>
> Our setting is fundamentally different from semi-supervised learning. In semi-supervised learning, the unlabeled samples are real data drawn from the true distribution, whereas in our case the synthetic samples are model-generated. Moreover, our focus is on how an external verifier influences the recursive self-training estimator, a phenomenon that classical semi-supervised theory does not capture.
>
> ### **W3. FID on MNIST and Domain-Appropriate Embeddings.**
>
> We agree that the standard FID was originally proposed for natural images, and its embeddings may not perfectly capture perceptual similarity on MNIST. However, to ensure comparability with prior works, we followed a common practice in the literature and reported standard FID as done in [6,7,8], where standard FID has also been used on MNIST and shown it still correlates well with image quality.
>
> That said, we agree with the reviewer that domain-specific embeddings can provide a more meaningful evaluation. Following this suggestion, we trained a classifier from scratch, and took the last layer for the embedding. **The new experimental results have been added to Appendix D.3 in the updated manuscript.** The results confirm the same trend reported in Figure 3: verifier-filtered retraining consistently reduces FID during early rounds and then plateaus, in line with our theoretical predictions. This additional analysis demonstrates that our conclusions are robust to the choice of embedding.  **We also include qualitative results that closely match the quantitative FID trends in Appendix D.4.**
>
>
>
>
> ### **W4. Meaning of “Verifier Training Size” in Figure 3b.**
>
>
> Figure 3b examines how the verifier quality affects retraining. The “verifier training size” refers to the number of real samples used to train the verifier. A larger training size corresponds to a higher-quality verifier. As expected, stronger verifiers, those trained on more real data, produce greater FID improvements during retraining, while weaker verifiers (trained on fewer real samples) can even degrade performance.
>
> References:
>
> [6] Dai, B. and Wipf, D. Diagnosing and Enhancing VAE Models. *ICLR* 2019.
>
> [7] Leontev, Mikhail, et al. "Quality metrics of variational autoencoders." *2020 International Conference on Information Technology and Nanotechnology (ITNT)*. IEEE, 2020.
>
> [8] Chan, Derrick Adrian, and Siphesihle Philezwini Sithungu. "Evaluating the suitability of inception score and fréchet inception distance as metrics for quality and diversity in image generation." *Proceedings of the 2024 7th International Conference on Computational Intelligence and Intelligent Systems*, 2024.

---

> > ### Author Response · Authors · 2025-11-21
> >
> > ### **Q1. Clarification of the $\Omega$ Notation**
> >
> > Thank you for pointing this out. You are correct that the notation in the manuscript was inaccurate. Our intention was to use $\Omega(\sqrt{n_0})$, rather than the $\omega$ symbol. The correct assumption should be  $\lambda_j(X_0) = \Omega(\sqrt{n_0})$,  which means that there exists a constant $c > 0$ such that, for all sufficiently large $n_0$, $\lambda_j(X_0) \ge c \sqrt{n_0}.$
> >
> > Thus, we only require that each eigenvalue is bounded below by a constant multiple of $\sqrt{n_0}$; we do not assume it is given by an explicit function of $n_0$. We corrected this notation in the revised manuscript.
> >
> > **Why is this a reasonable assumption?** The condition $\lambda_j(X_0) = \Omega(\sqrt{n_0})$ is mild and holds whenever the rows of $X_0$ have a non-degenerate covariance. If the rows of $X_0$ are i.i.d. mean-zero with non-degenerate covariance $\Sigma$, then $\tfrac{1}{n_0} X_0^\top X_0 \to \Sigma$. Then $$
> > \lambda_j\left(\tfrac{1}{n_0} X_0^\top X_0\right) \to \lambda_j(\Sigma)
> > \quad\Rightarrow\quad
> > \lambda_j(X_0^\top X_0) = \Theta(n_0),
> > $$
> > and the corresponding singular value satisfies
> > $$
> > \lambda_j(X_0) = \sqrt{\lambda_j(X_0^\top X_0)} = \Omega(\sqrt{n_0}).
> > $$
> > Thus the condition $\lambda_j(X_0) = \Omega(\sqrt{n_0})$ is automatically satisfied under any non-degenerate covariance model.
> >
> >
> > ### **Q2. Takeaway of Theorem 3.1 and Equation (9)**
> >
> > We clarify the intuition behind Theorem 3.1 and the discussion surrounding Equation (9). Theorem 3.1 gives a closed-form MSE bound for the one-step verifier-filtered retraining estimator $\hat{\theta}^1$ and Equation (9) describes the MSE of the initial estimator $\hat{\theta}^0$ trained only on real data.
> >
> > Theorem 3.1 makes explicit how the error of $\hat{\theta}^1$ depends on three components:
> > - the **verifier bias** (captured by $m_{1j}$ and $m_{3j}$),
> > - the **verifier range/strictness** (captured by $m_{2j}$), and
> > - the **synthetic sample size** $n_1$.
> >
> > By comparing this expression with the real-data error in Equation (9), we can determine **when** verifier-filtered retraining reduces the loss and **by how much**. The verifier is beneficial when it reduces the bias (through smaller $m_{1j}$ and $m_{3j}$) and ensures that the synthetic variance term decreases when the synthetic sample size($n_1$) is large.
> >
> > To make this clearer, we additionally plot both the theoretical loss predicted by Theorem 3.1 and the empirical loss landscape obtained from simulations in the **updated manuscript (Figure 1)**. The two surfaces align closely. This shows that Theorem 3.1 not only explains the qualitative behavior but also quantitatively matches the empirical results.
> >
> > ### **Q3. Why Does Verification Variance Persist with a Perfect Verifier?**
> >
> > The bias term will be 0 but the variance term will still exist. This is because the verifier’s decision rule is built from the initial real data estimator \hat \theta_0, which is itself based on a finite amount of real data. As a result, the next-round \hat \theta_1 will not be deterministic. Its value fluctuates with the randomness in \hat \thata_0. This inherent randomness produces the variance term m_2j​, which does not disappear even with infinitely many synthetic samples in a single iteration. Consequently, multiple iterations are required for convergence. **In short: perfect verification eliminates bias, but it cannot eliminate the uncertainty inherited from the finite real data in $\hat\theta_0$.**
> >
> > ### **Q4. Interpretation of the Three Verifier Quality Phases.**
> >
> > We agree that it is intuitive that an accurate verifier should help, while a badly mis-specified verifier can hurt. Our contribution is to formalize what actually happens in the long-run dynamics of iterative retraining.Theorem 4.1 shows that the update is a contraction toward the verifier center $\theta_c$. So we want to emphasize how verifier quality directly controls the long-run outcome of synthetic retraining. Beyond this intuitive interpretation, Theorem 4.1 also characterizes the convergence rate of the long-run dynamics, showing that the speed at which the estimator converges depends on the sample size and the selectiveness of the verifier.

---

### Author Response · Authors · 2025-12-03
**To AC and SAC: Rebuttal Summary about Our Paper**

We thank all reviewers for their thoughtful and encouraging feedback. In particular, we appreciate reviewers Lw2s, SkJA, j2Ts, and ENpW for their positive assessments (score 6, a strong initial rating), and reviewer TG7U for the constructive suggestion. We would like to clarify, however, that we differ from reviewer TG7U on the use of RLVR: we do not view RLVR as a universal remedy or an optimal solution for our problem.

In our response, we summarize the major points discussed during the rebuttal, including (i) the applicability of our theoretical insights to large-scale LLM experiments, (ii) additional robustness experiments we conducted, (iii) potential theoretical extensions, and (iv) clarifications that distinguish our contributions from related work. We have incorporated new details and results into the revised PDF (highlighted in blue). Below, we outline the most important updates made during the rebuttal.

### **1. Our theoretical conclusions generalize to large-scale LLM experiments** (to address reviewers SkJA, TG7U, and ENpW’s concerns)

During the rebuttal period, we conducted experiments on a large-scale news-summarization task using the `XSUM` dataset and the state of art LLM `SmolLM2-135M`. We compared retraining with **filtered synthetic data** versus **unfiltered synthetic data** following our theoretical setup. The results show that (1) filtering synthetic data consistently improves model performance, and (2) performance converges as the number of training iterations increases. In contrast, retraining on unfiltered synthetic data yields no measurable improvement. These findings demonstrate that our theoretical conclusions **generalize to large-scale datasets and contemporary LLMs**. Further details are provided in **Appendix D.6** of the updated PDF. This address reviewer *SkJA*, *TG7U*, and *ENpW*'s scalability/generalization concerns regarding our experimental results on MNIST dataset.

### **2. Why not use reinforcement learning with verified rewards (RLVR)**  (to address reviewer TG7U’s concern)

In summary, RLVR is not the right tool for the setting we study, and the verifier-based filtering mechanism we analyze is in fact the standard and widely adopted approach for our setting.

RLVR only works when reward signals are deterministic and verifiable—mainly math, coding, and similar tasks. Our setting, however, involves domains where no reliable scalar reward exists: pre-training (where rewards are undefined), alignment (which relies on comparisons rather than absolute rewards), and open-ended generative tasks (where scalar rewards are noisy and subjective). In these cases, RLVR is either inapplicable or unstable.

By contrast, verifier-based filtering (generate → verify → retrain) is already the popular, scalable, and practically successful paradigm used in major LLM pipelines—including frontier models such as DeepSeek-Coder. Therefore, focusing on verifier-based filtering is both well-motivated and aligned with real-world practice, whereas RLVR simply does not generalize to the broad class of scenarios our work addresses.


### **3.Relaxing Theory Assumptions: Random Synthetic Covariates Designs and General Verifier Knowledge Shape**  (to address reviewer j2Ts and reviewer EnpW’s concern)

We clarified that two stylized assumptions used for analytic tractability: (i) orthogonal synthetic covariates and (ii) spherical verifier knowledge are not essential to our conclusions. While full proofs are less tractable in the broader setting, we added explanations and supporting experiments showing our contraction/convergence claims are robust.

In **Appendix D.1**, we provide simulation results under fully random covariate design demonstrating that  the convergence  behavior persists. We discussed replacing spherical knowledge sets with general convex sets, outlined why the mechanism still applies, and included two empirical examples supporting that the convergence trend holds under these broader shapes in **Appendix D.2**. These additions indicate our findings are not artifacts of the stylized assumptions; the phenomena persist in more general (though less tractable) regimes, as shown by the new experiments and clarifications.

---

> ### Author Response · Authors · 2025-12-03
>
> ### **4. Added experiments with different initial real-data sample sizes** (to address reviewers SkJA and ENpW’s concern)
>
> We added experiments using initial real-data sample sizes ranging from 1k to 60k. Our theory predicts that verifier-based retraining improves the model only when (1) the verifier contains external knowledge not available to the generator, and (2) this knowledge has sufficiently small bias to yield a favorable bias–variance trade-off. This is why the main paper focuses on small initial sample sizes, where the verifier has sufficiently more information-external knowledge-about the data distribution than the generator such that model improvement can actually be observed.
>
> Across different initial sample sizes, we observe varying degrees of improvement from verifier-based retraining, and in all cases FID eventually converges. When the generator is initialized on all 60k real images, verifier filtering no longer improves FID over the initial model (since the verifier has no external knowledge compared to the generator in this case), but it still prevents the severe collapse seen under unfiltered retraining, consistent with our theoretical predictions. Further details are provided in **Appendix D.5**
>
> ### **5. Other minor concerns raised by reviewers Lw2s & SkjA**
>
> To address reviewer Lw2s’s concern about our FID computation, we conducted additional experiments using a domain-appropriate embedding FID metric. The results remain consistent with those reported using the standard FID. Full details are provided in Appendix D.3 and Appendix D.4 of the updated manuscript.
>
> To address reviewer Lw2s’s concern regarding the distinction from semi-supervised learning, we clarify that our setting is fundamentally different: semi-supervised learning assumes access to real unlabeled samples drawn from the true data distribution, whereas our framework operates on model-generated synthetic data, which follows a distribution induced by the generator rather than the ground truth.
>
> To address reviewer SkjA’s concern regarding our connection to preference maximization, we emphasize that our work tackles a different theoretical question: preference-maximization frameworks curate data via a reward model that has no external knowledge of the data-generating process or the source of improvement, whereas our verifier explicitly leverages such external knowledge to guide selection Moreover, we provide finite-sample guarantees and iterative (long-run) analysis, which explicitly characterize the convergence trajectory—unlike prior methods that evaluate only population-level expected objectives. In addition to these main distinctions, our modeling assumptions and focus also differ in several minor ways. A detailed discussion is provided on page 2 of the updated manuscript.
>
> Given the above summary of our major responses and revisions, we believe we have addressed all reviewer concerns. We respectfully hope for positive consideration from the Area Chair and believe that our paper will be a good addition to the community.

---

### Meta-Review · Area_Chair_ogtq · 2026-01-06

**Summary:**

This paper presents a solid and rigorous theoretical study of verifier-guided synthetic retraining. Although the analysis is carried out in a simplified linear regression setting, the results are technically sound and well supported. The authors substantially addressed reviewer concerns in the rebuttal by adding new experiments and clarifying their relationship to existing work. Overall, I view this as a borderline paper but lean toward acceptance given the broadly positive feedback (4 out of 5 reviewers gave positive feedback). In the revised version, the authors should clearly state the scope and limitations of their approach, add the comparison with alternatives such as RLVR, and include the additional LLM experimental results as discussed in the rebuttal.

**Reviewer Concerns:**

Reviewer Lw2s's concerns are mostly addressed. The outstanding issue is the gap between theory on linear regression versus general generative modeling.

Reviewer SkJA's concerns are mostly addressed. The outstanding issue is the concern of expeirmental setup.

Reviewer j2Ts's concerns are addressed to a reasonable extent. The outstanding issues are the cases of overparameterization and evolving verifier.

Reviewer TG7U's concerns are addressed only partially. The remaining concerns include the toy setting, limited comparision with RLVR. The authors added a LLM summarization experiment, which may alleviate the reviewer TG7U's concerns.

Reviewer ENpW's concerns are almost addressed.

**Reviewer Scores:**

The reviewer TG7U might be able to increase the score slightly given the new experiments, but I am not sure about this since there is no evidence of increasing scores. The other reviewers would not be able to increase scores since they are all concerned about the limited scope of linear regression.

---

### Decision · Program_Chairs · 2026-01-26

Accept (Poster)